# Targeting vascular dementia: Molecular docking and dynamics of natural ligands against neuroprotective proteins

Zhizhong Wang[1,3], Sen Xu[2], Ailong Lin[3], Chunxian Wei[3], Zhiyong Li[3], Yingchun Chen[3], Bizhou Bie[3]*, Ling Liu[4]*

1 College of Chinese Medicine, Hubei University of Chinese Medicine, Wuhan, Hubei, China, 2 School of Medicine, Wuhan University of Science and Technology, Wuhan, China, 3 The Third People's Hospital of Hubei Province, Affiliated Jianghan University, Wuhan, Hubei, China, 4 Hubei Provincial Hospital of Traditional Chinese Medicine, Affiliated Hospital of Hubei University of Chinese Medicine, Wuhan, China

* 18771032136@163.com (BB); 18971537282@163.com (LL)

## Abstract

Vascular dementia (VaD), a neurodegenerative disease driven by vascular pathology, requires multi-targeted therapeutic strategies. This study employs an integrated in silico approach to evaluate the neuroprotective potential of natural ligands against key proteins implicated in VaD pathogenesis. Using molecular docking and normal mode analysis (NMA), four natural compounds (Galangin, Resveratrol, Curcumin, and Licocumarone) were assessed for their binding affinity and structural influence on six target proteins: APLP1, APOE, CLDN5, SOD1, MMP9, and MTHFR. Docking analysis revealed that galangin exhibited the highest binding affinity to APLP1 (−8.5 kcal/mol), resveratrol to MTHFR (−8.1 kcal/mol), and curcumin showed dual efficacy toward APOE (−7.2 kcal/mol) and MMP9 (−8.0 kcal/mol). Licocumarone demonstrated notable stabilization of CLDN5 and SOD1. The NMA results indicated ligand-induced stabilization of protein cores and enhanced flexibility in loop regions, which may impact amyloid aggregation, oxidative stress, and blood-brain barrier integrity. Pathway enrichment using the KEGG and Reactome databases identified significant involvement of the IL-17 and TNF signaling pathways, along with leukocyte transendothelial migration, linking inflammation with vascular dysfunction. APOE emerged as a central node within the protein-protein interaction network, highlighting its regulatory importance. This study highlights the therapeutic relevance of natural ligands as cost-effective modulators of multiple VaD-associated pathways. The combined use of molecular docking, protein dynamics, and enrichment analyses provides a comprehensive computational framework for early-stage drug discovery. These findings warrant further experimental validation to advance the development of targeted, mechanism-driven interventions for vascular dementia.

**Data availability statement:** All relevant data are within the paper and its Supporting Information files.

**Funding:** This work was funded by the Project of Hubei Provincial Administration of Traditional Chinese Medicine (grant ZY2025Q007).

**Competing interests:** The authors have declared that no competing interests exist.

# 1. Overview of Vascular Dementia (VaD)

Vascular dementia (VaD) is characterized by cerebral infarcts, myelin loss, and white matter lesions that are usually associated with amyloid angiopathy. These vascular lesions result in neuronal loss and synaptic breakdown and have a profound effect on cognitive processes [1]. Its pathophysiology is attributed to chronic cerebral hypoperfusion, which interferes with autophagy and results in hippocampal and neuronal loss [1]. In addition, endothelial dysfunction and decreased nitric oxide bioavailability are responsible for vascular disease, which is a characteristic of VaD [2]. Oxidative stress is significant in VaD pathogenesis, with an imbalance between pro-oxidant and anti-oxidant species resulting in increased levels of reactive oxygen species (ROS) [2,3]. Endothelial dysfunction and vascular diseases result from an oxidative environment. Inflammation is another important factor that disrupts neuroinflammation and apoptosis and causes damage to endothelial and neuronal cells [1]. Endothelial function is also impaired by oxidative stress, which can influence amyloid precursor protein processing, resulting in amyloid-beta accumulation in the blood vessels of the brain [4].

Furthermore, Apolipoprotein E (ApoE) is a late-onset genetic risk factor for dementia, which is strongly associated with Alzheimer's disease. The ApoE epsilon 4 allele is related to elevated beta-amyloid deposition in the cerebral cortex, which promotes vascular and [2] neurodegenerative pathologies [5–7]. The processing of amyloid precursor protein (APP) is subject to oxidative stress, resulting in the enhanced secretion of amyloid-beta peptides. It is involved in the amyloidogenic pathway in the pathogenesis of vascular diseases and in amyloid deposition in the brain. Claudin-5 is a tight junction protein involved in blood-brain barrier integrity. Its malfunction results in increased permeability and is implicated in the pathogenesis of vascular dementia by causing harmful substances to penetrate the brain [8]. Superoxide dismutase (SOD) is an antioxidant enzyme that prevents oxidative stress by catalyzing the conversion of superoxide radicals into oxygen and hydrogen peroxide. Its function in vascular dementia includes the prevention of endothelial cell damage by oxidation and preservation of vascular function. Matrix metalloproteinase-9 (MMP-9) is involved in the degradation of extracellular matrix components. MMP-9 overexpression may lead to disruption of the blood-brain barrier, vascular injury, and neuroinflammation in dementia [1]. Methylenetetrahydrofolate reductase (MTHFR) is an enzyme that is involved in homocysteine metabolism. MTHFR mutations have the potential to result in hyperhomocysteinemia, endothelial dysfunction, and oxidative stress risk factors for vascular diseases and dementia [3]. Therefore, targeted proteins, such as ApoE and APP, play significant roles in the progression of vascular dementia, highlighting potential therapeutic targets for intervention. Table 1 summarizes the different proteins and their roles in vascular dementia.

We chose four natural ligands for this study because natural substances are known to have fewer side effects and are more cost effective. Curcumin, a polyphenolic molecule extracted from turmeric, exhibits abundant anti-inflammatory and antioxidant activities. Curcumin has been extensively studied for its ability to prevent amyloid protein aggregation, a feature of neurodegenerative diseases, such as Alzheimer's disease [9–11]. Galangin is a flavonoid found in *Alpinia officinarum* and

**Table 1. The roles of different proteins in vascular dementia.**

| Protein | Main Function | Role in VaD |
|---------|---------------|-------------|
| APOE | Lipid metabolism, BBB maintenance | APOE4 variant increases the risk of VaD via neurovascular dysfunction |
| APLP1 | Amyloid processing, synaptic function | Affects cerebrovascular amyloid deposition, leading to vascular pathology |
| CLDN5 | Maintains BBB integrity | Loss leads to vascular leakage, neuroinflammation, and cognitive decline |
| SOD1 | Antioxidant defense | Reduced SOD activity leads to oxidative stress-induced neurovascular damage |
| MMP9 | Extracellular matrix remodeling | High levels contribute to BBB breakdown, white matter lesions, and vascular injury |
| MTHFR | Homocysteine metabolism | Mutations lead to vascular dysfunction, increasing VaD risk |

in other plants. Galangin possesses antioxidant and anti-inflammatory properties that have been implicated in neuroprotection. Nonetheless, Licocumarone, a derivative of coumarin, has been shown to exhibit therapeutic activities, such as anti-inflammatory and antioxidant activities, and is known to help treat neurodegenerative disorders. Moreover, Resveratrol is a polyphenolic compound found in red wine and certain plants. It has been found to induce non-amyloidogenic pathways, enhance amyloid-beta clearance, and reduce neuronal damage, thereby emerging as a potential neuroprotective drug [12–14]. Curcumin has shown great promise in reducing amyloid burden and rescuing neuronal injury in several models of neurodegenerative diseases, such as Alzheimer's and Parkinson's diseases [9–11]. Resveratrol has been found to increase the clearance of amyloid-beta peptides and reduce neuronal injury, which are important in the management of Alzheimer's disease [12–14]. Galangin and Licocumarone have been studied in detail because their antioxidant and anti-inflammatory activities imply their possible neuroprotective functions.

The study's major aim is to explore the interactions of selected proteins with natural ligands such as curcumin, galangin, licocumarone, and resveratrol using molecular docking. This can help determine the possible mechanisms through which natural compounds exert neuroprotective actions, specifically in the context of vascular dementia [15]. Molecular docking experiments can shed light on the binding affinities and binding sites of these ligands towards target proteins and can lead to the identification of new therapeutic avenues for neurodegenerative diseases [16,17].

Research combining normal mode analysis (NMA) and molecular docking in the context of VaD therapies is lacking. NMA offers information about the inherent flexibility and collective movement of proteins, which can have a substantial impact on ligand binding and drug efficacy. Conversely, molecular docking is commonly employed to predict the binding and orientation of small molecules to protein targets [18]. By considering protein dynamics throughout the docking process, the combination of these two approaches may provide a more thorough understanding of protein-ligand interactions. However, the use of NMA in conjunction with docking has not been documented in the literature for VaD therapies [19]. Previous research has concentrated on static docking techniques and, occasionally, molecular dynamics simulations, but has not taken advantage of NMA's special benefits of NMA to investigate the potential effects of protein conformational changes on drug binding and therapeutic results. This is a major knowledge gap because combining NMA and docking should improve the predictive ability of computational drug discovery for VaD and possibly result in the discovery of more potent treatment candidates [20].

## Materials and methods

### 1) Selection of signaling proteins

The selection of these proteins, namely Apolipoprotein E (APOE), amyloid beta precursor protein (APLP1), Claudin-5 (CLDN5), Superoxide Dismutase (SOD1), Matrix Metalloproteinase-9 (MMP-9), and Methylenetetrahydrofolate Reductase (MTHFR), was based on KEGG pathway analysis, enabling the selection of key proteins that directly hold mechanisms and regulatory roles in vascular dementia (VaD) [21]. These proteins were selected because they are implicated in crucial pathways such as neuroinflammation, blood-brain barrier integrity, oxidative stress, amyloid

metabolism, and homocysteine regulation. These pathways have been proposed to be involved in the pathogenesis of vascular dementia [22].

Although several genes were enriched by KEGG analysis, we chose APOE, APLP1, CLDN5, SOD1, MMP9, and MTHFR due to their repeated enrichment across multiple VaD-relevant pathways (e.g., IL-17, TNF, leukocyte transendo-thelial migration, oxidative stress), central roles in PPI networks, and functional conservation across species. Their roles in vascular integrity, amyloid metabolism, oxidative stress management, and neuroinflammation made them ideal candidates for structural modeling and pharmacological screening.

### 2) Investigating the Co-expression of genes

CLDN5, APOE, MMP9, APLP1, SOD1, and MTHFR were identified through a comprehensive literature analysis that emphasized their involvement in vascular dementia (VaD). Functional associations and co-expression relationships among genes/proteins across numerous organisms, such as human brain tissues and model organisms (Mus musculus, Danio rerio, and Gasterosteus aculeatus), were visualized using STRING Database (https://string-db.org/cgi/coexpression?taskId=b3Ei1T4T2&sessionId=bmZLtQmlVdxb&allnodes=1), version 12.0. Gene co-expression analysis was conducted using the STRING database. We implemented a minimum interaction score of 0.7 to ensure robust confidence in the interactions. STRING's "Gene Coexpression" visualization tool was employed to construct functional co-expression networks that illustrate interactions and functional modules that are pertinent to vascular well-being [23,24].

### 3) Protein-Protein interaction

PPI Network Analysis of the STRING database (version 12.0) investigated the Protein-Protein Interaction (PPI) network by utilizing a high-confidence criterion set at an interaction score of 0.7 to analyze genes (CLDN5, APOE, MMP9, APLP1, SOD1, and MTHFR) [25]. The Network Analyzer plugin was employed to observe and investigate the network generating centrality metrics, including degree, betweenness centrality, and clustering coefficients, in Cytoscape (version 3.10.0).

### 4) Gene set enrichment analysis (GSEA)

Gene set enrichment analysis (GSEA) was conducted using Enrichr, a bioinformatics tool frequently employed for functional annotation of gene lists. The Reactome pathways 2024 database and KEGG 2021 Human pathways were employed to analyze the target genes (CLDN5, APOE, MMP9, APLP1, SOD1, and MTHFR) [26]. Statistical significance was evaluated using p-values and modified q-values to determine the false discovery rate. Manhattan plots, UMAP clustering (utilizing the Leiden algorithm), and volcano plots were implemented to emphasize pathways that were exceedingly abundant in information. Enhanced pathways with high statistical significance and odds ratios were prioritized to identify significant biological processes associated with vascular dementia.

### 5) Protein sequence retrieval and modelling

Protein sequences were obtained from UniProt (https://www.uniprot.org/), an exhaustive and well-curated protein sequence database that provides elaborate annotations, functional insights, and structural information about proteins. Although some experimentally analyzed structures from the Protein Data Bank (PDB) (https://www.rcsb.org/) are available, we preferred modelling based on trRosetta (https://yanglab.qd.sdu.edu.cn/trRosetta/) to ensure full-length completeness. This led to consistent modeling across different proteins and circumvented variations under different experimental conditions [27]. The predicted models were also optimized for use in molecular docking, making them suitable for studying protein-ligand interactions in the context of vascular dementia.

## 6) Selection of natural ligands

The chemical structures of these natural ligands were determined using PubChem and ChEMBL databases, which provide detailed molecular information [28].

## 7) Active site prediction and molecular docking

The active site for each protein was determined using DoGSiteScorer (https://proteins.plus/), and molecular docking was conducted using AutoDock Vina, a popular tool for predicting how small molecules, such as drugs, bind to receptors with known 3D structures. AutoDock offers a collection of automated docking tools to predict ligand binding to macromolecular targets [19,29,30]. Each ligand was docked independently five times against its particular protein target with AutoDock Vina, allowing for random variance in initial conformations. The pose with the lowest binding energy (in kcal/mol) was chosen for further study, corresponding with the normal practice in molecular docking. To assess the consistency and variability of the docking process, we averaged the binding scores from all five runs per ligand-protein pair and calculated the standard deviation. Supplementary Tables 1 and 2 in S1 File summarizes these values, which validate the stability and reproducibility of binding predictions.

## 8) Scoring functions

The assessment of docking outcomes depends mainly on scoring functions that estimate the affinity and stability of ligand–receptor binding. The most important measures are the binding energy, which indicates the strength of the ligand-target affinity, and the inhibition constant, which is a measure of ligand potency. The presence of hydrogen bonds and hydrophobic interactions was also evaluated to determine the type of binding [19,29,30].

## 9) Normal mode analysis of the docked complex

Normal mode analysis (NMA) was performed using iMODS. It calculates molecular mobility and structural flexibility by including the coordinates of the docked complex. iMODS presents the deformability plot findings, eigenvalues, covariance matrix, and elastic network.

## 11) Molecular dynamics simulation and post-processing analysis

All protein-ligand complexes were subjected to 100 ns molecular dynamics simulations using (e.g., GROMACS) with NPT ensemble settings at 300 K and 1 atm pressure. Trajectory analyses were used to determine structural stability and conformational behavior. Root Mean Square Deviation (RMSD), Root Mean Square Fluctuation (RMSF), radius of gyration (Rg), and solvent-accessible surface area (SASA) were calculated using built-in GROMACS tools (gmx rmsd, gmx rmsf, gmx gyrate, gmx sasa) and visualized with xmgrace. These measurements were utilized to analyze the complexes' dynamic behavior and binding stability during the simulation period.

## 11) Visualization tools used

Docking results were analyzed and visualized using visualization tools such as Discovery Studio and Ligplot. These tools enable researchers to visually analyze binding interactions, evaluate ligand fit within the active site, and highlight the essential interactions that contribute to binding affinity. These visualizations are beneficial for assessing docking data and for directing further experimental validation [19,29,30].

## Results

### 1) Retrieval of signaling proteins

The chosen proteins Apolipoprotein E (APOE), Amyloid Beta Precursor-Like Protein (APLP), Claudin-5 (CLDN5), Superoxide Dismutase (SOD), Matrix Metalloproteinase-9 (MMP-9), and Methylenetetrahydrofolate Reductase (MTHFR) play a

key role in the progression of vascular dementia. Amyloid metabolism and neurodegeneration are closely associated with APOE and APLP.

APOE regulates amyloid-beta clearance; APLP governs neuronal signaling. A key characteristic of both Alzheimer's disease and vascular dementia is the accumulation of amyloid-beta, which arises from the dysfunction of these proteins. APOE interacts with the receptor for advanced glycation end products (RAGE), a key component of the neuroinflammatory pathways that exacerbate brain damage (Fig 1).

Claudin-5 (CLDN5) (Fig 2) and Matrix Metalloproteinase-9 (MMP-9) are essential for preserving the integrity of the blood-brain barrier (BBB). Tight junctions that regulate blood-brain barrier permeability mostly rely on CLDN5, whereas MMP-9 facilitates extracellular matrix remodeling [31]. Dysregulation of these proteins leads to disruption of the blood-brain barrier, increased permeability, and infiltration of neurotoxic substances into the brain, thereby accelerating the progression of vascular dementia. Senescence-associated secretory phenotype (SASP) MMP-9 induces inflammation and tissue alteration when elevated in the SASP (Fig 3). It facilitates inflammatory signals and tissue damage in vascular dementia, resulting in the degradation of the extracellular matrix (ECM).

The one-carbon metabolism pathway regulates homocysteine concentrations through MTHFR, a methylenetetrahydrofolate reductase. Increased homocysteine levels resulting from MTHFR deficiency are recognized as markers of vascular damage, endothelial impairment, and cognitive deterioration. Homocysteine-induced vascular damage contributes to the pathological development of vascular dementia by promoting neuroinflammation and reducing the cerebral blood flow.

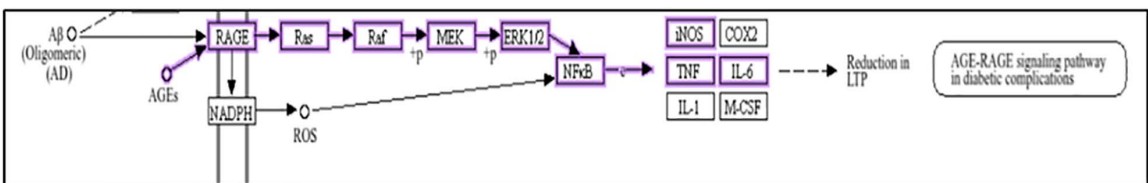

**Fig 1. Age-Rage Signaling pathway which involves Amyloid beta precursor-like protein.**

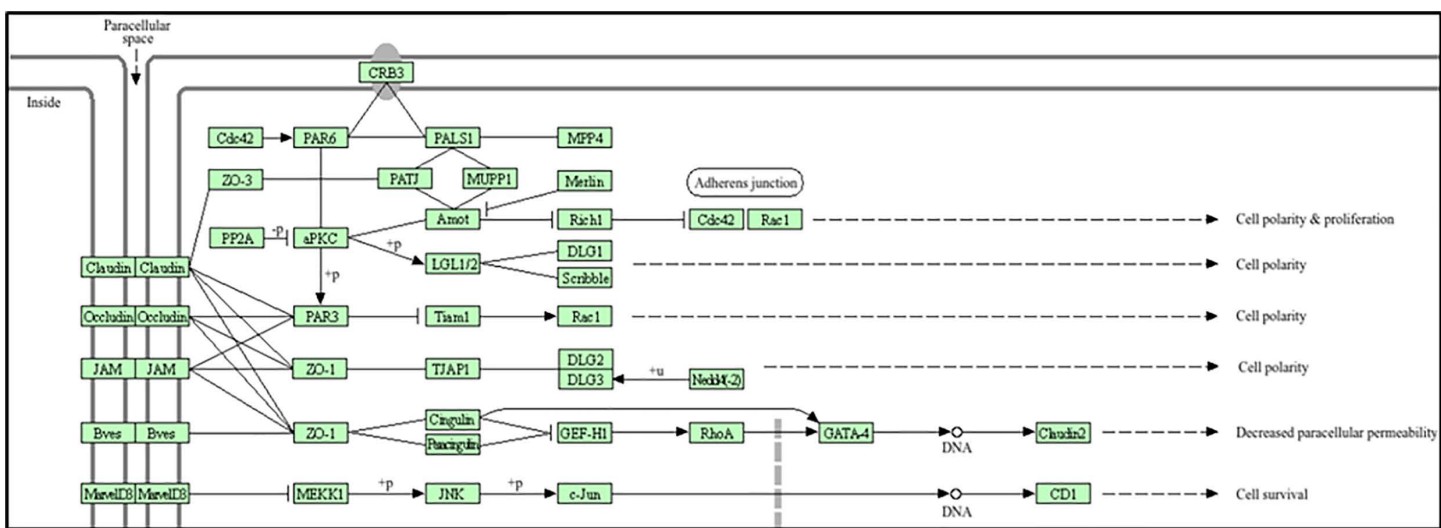

**Fig 2. Claudin-5 is involved in one of the main proteins that control the blood-brain barrier (BBB).** Higher permeability due to dysfunction lets harmful chemicals enter the brain leading to vascular dementia.

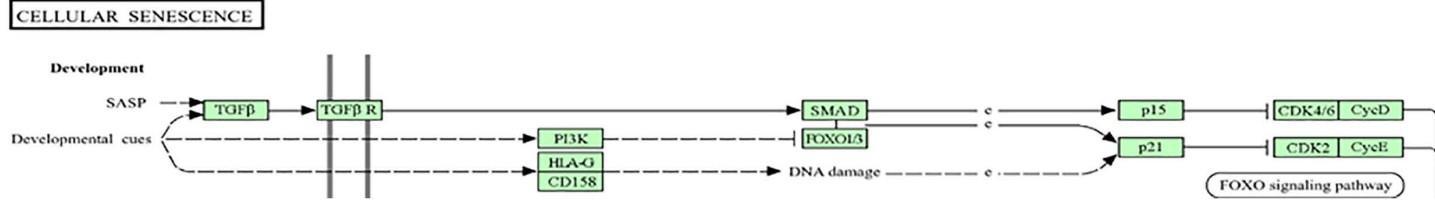

**Fig 3. Matrix metalloproteinase-9 is involved in the SASP signaling pathway.**

THF is a key folate metabolism intermediate [21]. It is crucial for DNA synthesis, methylation, and homocysteine stabilization. Brain function is dependent on Thymidylate Synthase (EC: 2.1.1.45), which ensures DNA replication and repair. It also allows 5-Methyl-THF to remethylate homocysteine to methionine via Methylenetetrahydrofolate Reductase (MTHFR; EC: 1.5.1.20, 1.5.1.53, and 1.5.1.54) (Fig 4). Hyperhomocysteinemia induces endothelial dysfunction, oxidative stress, and neuroinflammation if this pathway fails. This reduces the blood flow in the brain and accelerates neurodegeneration. It also modulates gene expression and histone methylation, thereby affecting neuronal survival and cognition. Since 5,10-Methylene-THF deficiency or breakdown can worsen vascular and neurological illnesses, it is crucial to vascular dementia.

Oxidative stress is a primary contributor to brain damage in vascular dementia, and Superoxide Dismutase (SOD1) plays a crucial role in detoxifying reactive oxygen species (ROS). Impaired SOD function leads to excessive ROS accumulation, resulting in oxidative damage, mitochondrial dysfunction, and ultimately neuronal death. Superoxide Dismutase

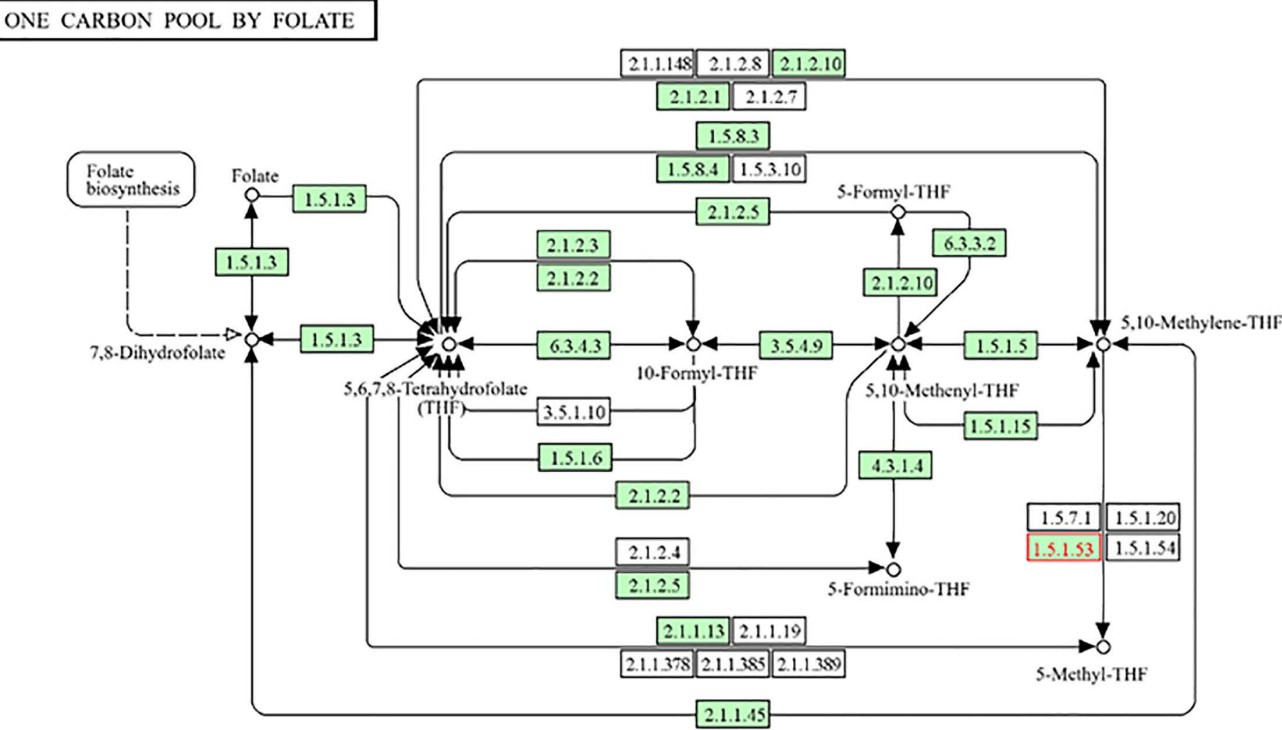

**Fig 4. Central Role of 5,10-Methylene-THF in DNA Synthesis, Methylation, and Homocysteine Regulation; One-Carbon Metabolism Pathway.**

(SOD1) is regulated by FOXO transcription factors in the FOXO signalling pathway (Fig 5). This increased the oxidative stress response. Mn-SOD reduces "mitochondrial oxidative stress" and converts superoxide radicals into harmless molecules, which contributes to vascular dementia [32,33]. Its absence causes (Blood-Brain Barrier) BBB issues, mitochondrial damage, and quicker cell death, which accelerates cognitive impairment in patients with VaD. Neuroinflammation and blood vessel damage result from chronic oxidative stress, outpacing SOD1 activity. Antioxidants and SIRT1 activators may decrease the progression of vascular dementia by repairing the oxidative damage.

The selection of proteins for docking investigations in vascular dementia is based on their connections with basic disease pathways. Targeting these proteins may provide therapeutic insights for this deadly disease. These proteins may be used to improve cognitive function in patients with vascular dementia and to delay their progression.

## 2) Gene Co-expression analysis

A study of gene co-expression showed that CLDN5, APOE, MMP9, APLP1, SOD1, and MTHFR are tightly regulated in the human brain tissue. The expression patterns of these genes are similar in model animals such as *Mus musculus*, *Danio rerio*, and *Gasterosteus aculeatus*. The high expression of these genes suggests that they work together to support the vascular well-being, oxidative stress response, and neuroinflammation pathways (Fig 6). These are important processes associated with vascular dementia (VaD). This demonstrates that dysregulation of one gene can affect the expression of other genes, which in turn supports the role of these genes in the pathophysiology of vasodilation disease (VaD). This finding matches the information already published that these proteins are linked to amyloid processing, changing the structure of cerebrovascular networks, and maintaining the blood-brain barrier.

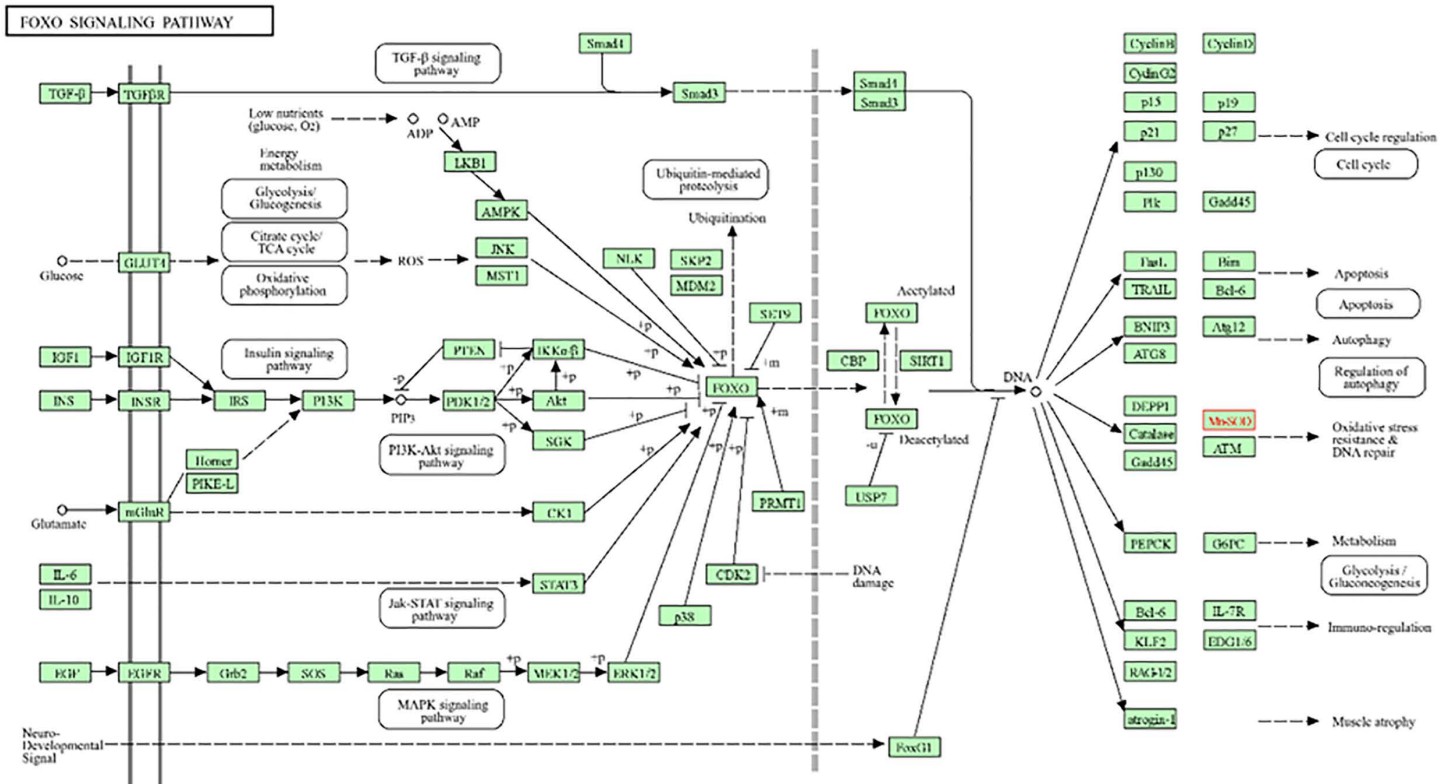

**Fig 5. Shows the role of the FOXO signaling pathway and SOD1 in vascular dementia.**

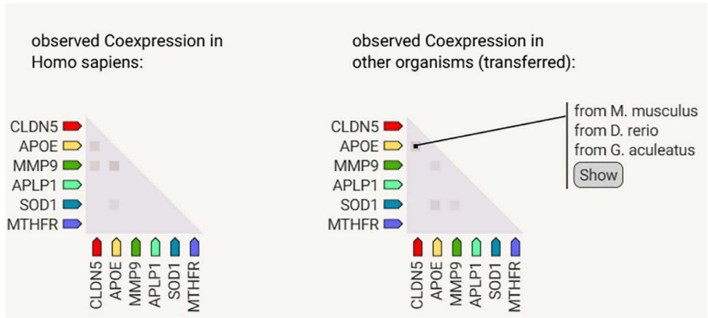

**Fig 6. These genes were found in humans (left) and model animals (right): CLDN5, APOE, MMP9, APLP1, SOD1, and MTHFR.** In the co-expression study of genes, the gene co-expression pattern suggested vascular and neurodegenerative processes due to their functional connections.

All these factors are fundamental to the pathology of vascular dementia (VaD) pathology. Functional enrichment analysis of CLDN5, APOE, MMP9, APLP1, SOD1, and MTHFR revealed their significant association with neurovascular dysfunction and inflammation. Previous studies of co-expression revealed a strong functional link among these genes; the current enrichment analysis validated their shared participation in cerebrovascular control (Fig 7). The FDR color gradient (from light green to dark blue) reflects the statistical relevance, with lower FDR values suggesting greater enrichment. Furthermore, the bubble size denotes the number of genes linked to each pathway. These results indicated the relevance of these genes as possible hereditary risk factors and therapeutic targets, thereby strengthening their role in VaD.

### 3) Protein-protein interaction network

PPI network analysis revealed strong functional relationships between CLDN5, APOE, MMP9, APLP1, SOD1, and MTHFR, and vascular integrity, oxidative stress, neuroinflammation, and amyloid metabolism. NOS3 and APOE control

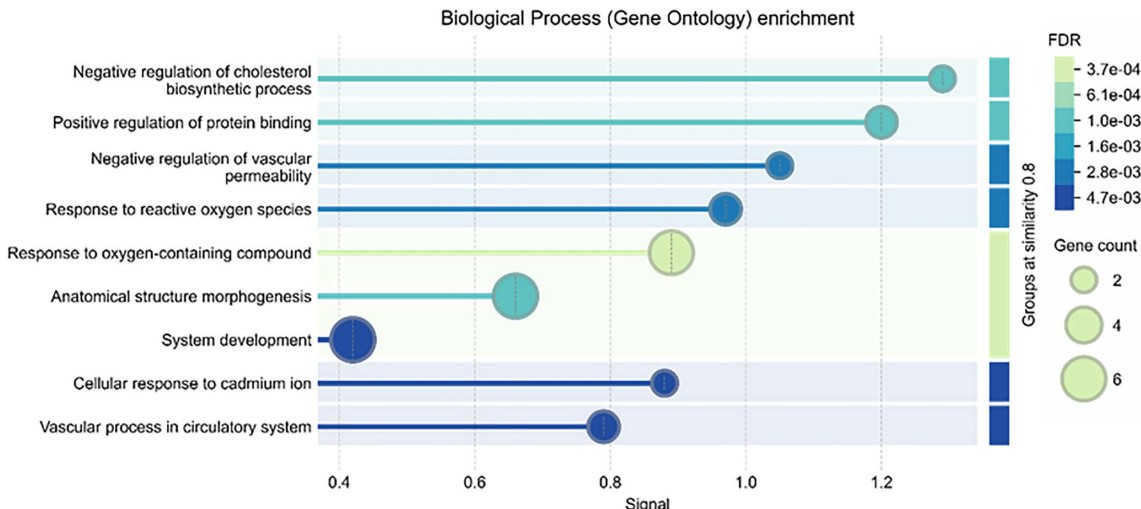

**Fig 7. Functional enrichment study of CLDN5, APOE, MMP9, APLP1, SOD1, and MTHFR revealed notably enhanced biological processes associated with blood-brain barrier integrity, oxidative stress, and neurovascular dysfunction.** While the color gradient shows the false discovery rate (FDR), with darker tones denoting greater significance, bubble size indicates the number of related genes.

blood-brain barrier (BBB) integrity and vascular function, whereas CLDN5 and MMP9 maintain the BBV and remodel the extracellular matrix. SOD1 and MTHFR are linked to oxidative stress and endothelial dysfunction, indicating their role in neurovascular injury. This network is related to the pathogenesis of dementia by APLP1 and MMEL1, which help amyloid metabolism and neurodegeneration. Their biological relevance to vascular dementia and therapeutic potential are shown in (Fig 8).

Protein-protein interaction (PPI) network analysis revealed that APOE acts as a major hub connecting several key pathways involved in vascular dementia (VaD). As shown in Table 2, it exhibits the highest degree (8) and betweenness centrality (0.32), reflecting its central regulatory role. CLDN5 and SOD1 had high clustering coefficients (1.5 and 1.8, respectively), suggesting their involvement in tightly coordinated functional modules. MMP9 and MTHFR, while not as

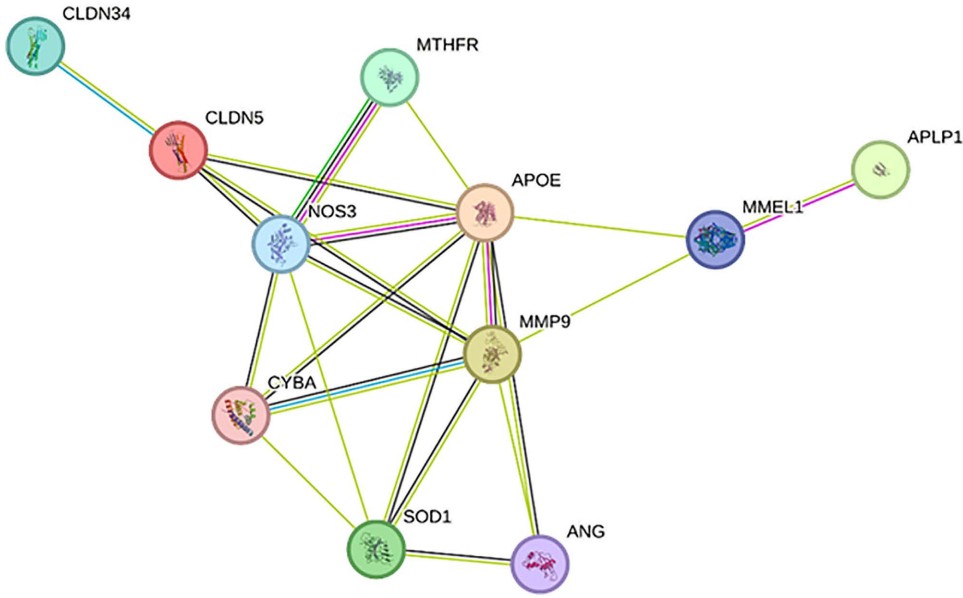

**Fig 8. The protein-protein interaction (PPI) network of CLDN5, APOE, MMP9, APLP1, SOD1, and MTHFR indicated their functional relationship with vascular integrity, oxidative stress, neuroinflammation, and amyloid metabolism.** NOS3 and APOE are key hubs for blood-brain barrier regulation and vascular dysfunction. The colored edges indicate many functional interactions, indicating the role of these proteins in the pathogenesis of vascular dementia.

**Table 2. Network metrics for Protein-Protein Interaction (PPI). The table shows important network measures for certain proteins associated with vascular dementia. Degree shows how many links each protein has, Betweenness Centrality shows how it acts as a network hub, Clustering Coefficient shows how connected its neighbours are, and Edge Confidence Score shows how reliable the interaction is based on estimates from the STRING database.**

| Protein | Degree | Betweenness Centrality | Clustering Coefficient | Edge Confidence Scores |
|---------|--------|------------------------|------------------------|------------------------|
| APLP1 | 1 | 0.30 | 1.00 | 0.597 |
| APOE | 8 | 0.32 | 1.42 | 0.777 |
| CLDN5 | 4 | 0.2 | 1.5 | 0.756 |
| SOD1 | 5 | 0.10 | 1.8 | 0.631 |
| MMP9 | 7 | 0.20 | 1.52 | 0.733 |
| MTHFR | 2 | 0.20 | 1.0 | 0.843 |

central as APOE, displayed strong connectivity and high edge confidence scores (>0.7), indicating their reliable interactions with other critical proteins.

The edge confidence scores from STRING reflect the integrated probability of functional associations between protein pairs. In our network, APOE showed well-supported connections with CLDN5 (0.756) and MMP9 (0.733), linking lipid metabolism, vascular integrity, and inflammatory pathways. MTHFR had the highest confidence score (0.843), emphasizing its specific role in homocysteine metabolism and endothelial dysfunction. SOD1, with a moderate score (0.631), is associated with the oxidative stress response and vascular damage, and its clustering suggests tight functional integration. CLDN5 and MMP9, both essential for blood-brain barrier (BBB) regulation and extracellular matrix remodeling, also showed strong interactions, reinforcing their cooperative involvement in the BBB disruption seen in VaD. Although APLP1 had the lowest degree (1) and confidence score (0.597), its relevance to amyloid processing pathways supports its inclusion as a specialized node in the network. Overall, these interaction metrics validated the biological significance of the selected proteins and supported their collective contribution to the molecular framework underlying vascular dementia.

## 4) Gene set enrichment analysis (GSEA)

GSEA was performed using Enrichr, a powerful gene set enrichment analysis (GSEA) application that enables the analysis of lists of genes or proteins based on predominant biological pathways, molecular functions, and disease associations. It is widely used for functional annotation in bioinformatic and systems biology studies. KEGG 2021 and Reactome pathways 2024 were analyzed for the targeted gene set.

The KEGG pathway enrichment study (Fig 9) revealed many important biological processes that may play a role in the pathogenesis of vascular dementia. It involves CLDN5 and MMP9. Leukocyte transendothelial migration (p = 0.000476, q = 0.013326) was the most important process. Evidence suggests that folate (MTHFR, p = 0.005986) is associated with homocysteine metabolism and vascular failure. Other pathways worth investigating include cholesterol metabolism

**Table of top 10 significant p-values and q-values for KEGG 2021 Human**

| term | p-value | q-value | overlap_genes |
|---|---|---|---|
| Leukocyte transendothelial migration | 0.000476 | 0.013326 | [CLDN5, MMP9] |
| One carbon pool by folate | 0.005986 | 0.083731 | [MTHFR] |
| Bladder cancer | 0.012239 | 0.083731 | [MMP9] |
| Cholesterol metabolism | 0.014908 | 0.083731 | [APOE] |
| Peroxisome | 0.024352 | 0.083731 | [SOD1] |
| IL-17 signaling pathway | 0.027874 | 0.083731 | [MMP9] |
| Prostate cancer | 0.028753 | 0.083731 | [MMP9] |
| Longevity regulating pathway | 0.030216 | 0.083731 | [SOD1] |
| TNF signaling pathway | 0.033137 | 0.083731 | [MMP9] |
| Relaxin signaling pathway | 0.038086 | 0.083731 | [MMP9] |

**Fig 9. KEGG 2021 human enrichment study identified the top 10 most important pathways.** The figure shows the biological paths with the most enriched p-values, q-values, and matched overlapping genes. The p-value shows statistical significance, whereas the q-value corrects for repeated testing errors (false discovery rate). Key pathways include leukocyte transendothelial movement, lipid metabolism, IL-17 signaling, and TNF signaling. Genes such as MMP9, CLDN5, APOE, and SOD1 control these pathways.

(APOE, p = 0.014908), which is consistent with the role of APOE in lipid metabolism and neurodegeneration. Peroxisomes and processes that control aging (SOD1) also suggest a possible role for oxidative stress in the development of disease.

These immune-related pathways, including IL-17 signaling (p = 0.027874) and TNF signaling (p = 0.033137), both involving MMP9, demonstrated the role of neuroinflammation in the development of vascular dementia. MMP9 plays many biological roles, such as changing the extracellular matrix and the blood-brain barrier. It is involved in many processes, such as bladder cancer, prostate cancer, and relaxin signaling.

The Manhattan plot (Fig 10) shows how statistically important enhanced KEGG pathways are based on gene set enrichment research. On the y-axis, each data point shows a KEGG pathway along with its -log$_{10}$ (p-value). Points that are higher show lines that are more strongly linked to the list of input genes because they have a stronger enrichment. A clear peak indicates that at least one route is very important and should be studied further. On the other hand, tracks with lower y-axis values are not as important, but might still be a part of how the disease works in general.

Many pathways in more advanced stages are likely to be associated with vascular dementia. These include the factors that control cholesterol levels, neuroinflammation, oxidative stress, and endothelial failure. Since their association with neurodegeneration and vascular damage. When this enrichment method is combined with functional validation, it helps identify the most statistically significant pathways for therapeutic targeting and biomarker identification in vascular dementia studies.

The UMAP-based plot (Fig 11) provides a clear picture of the improved pathways discovered by KEGG 2021 Human analysis, which allowed us to identify the functionally linked biological processes. The Leiden clustering approach groups pathways with linked functional goals, with each point representing a KEGG pathway. The larger black points show greatly enhanced paths, emphasizing their biological importance in the dataset. Important pathways in vascular dementia, such as leukocyte transendothelial movement, cholesterol metabolism, IL-17 signaling, and TNF signaling, are likely to be grouped in different ways to show how they work at the molecular level. For instance, vascular dysfunction-related pathways (such as leukocyte transendothelial migration and peroxisome regulation) can be grouped, indicating their role in maintaining endothelial integrity and controlling the blood-brain barrier. Neuroinflammation-related pathways (such as IL-17 and TNF signaling) were grouped, indicating the role of immune-mediated processes in vascular dementia. Furthermore, metabolic pathways, such as cholesterol metabolism and the one-carbon pool by folate, constitute another cluster and contribute to metabolic inputs in neurovascular dysfunction.

Reactome pathway enrichment analysis (Fig 12) identified vascular dementia-related biological processes by emphasizing lipoprotein, vitamin, and endothelial function. APOE supports cholesterol metabolism and neurovascular health by

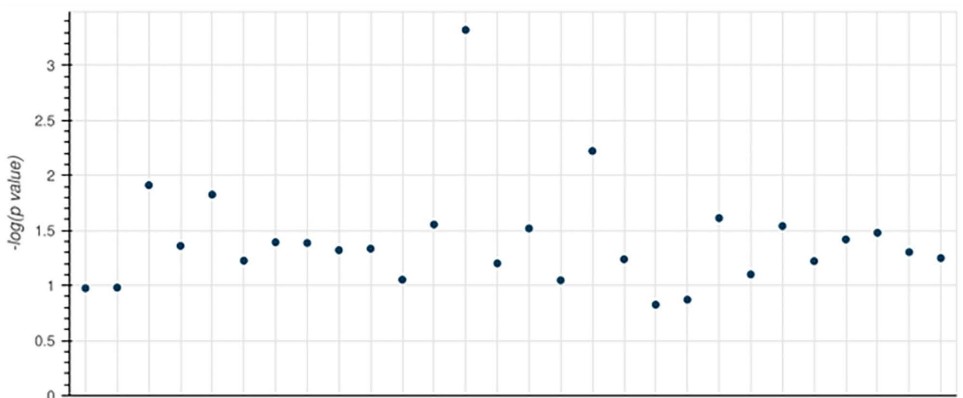

**Fig 10. Manhattan map of KEGG pathway enrichment.** The enrichment significance is shown on the y-axis, with each point on the x-axis representing a specific KEGG pathway. The statistical importance increased with the y-axis values, and the peaks indicated the best pathways.

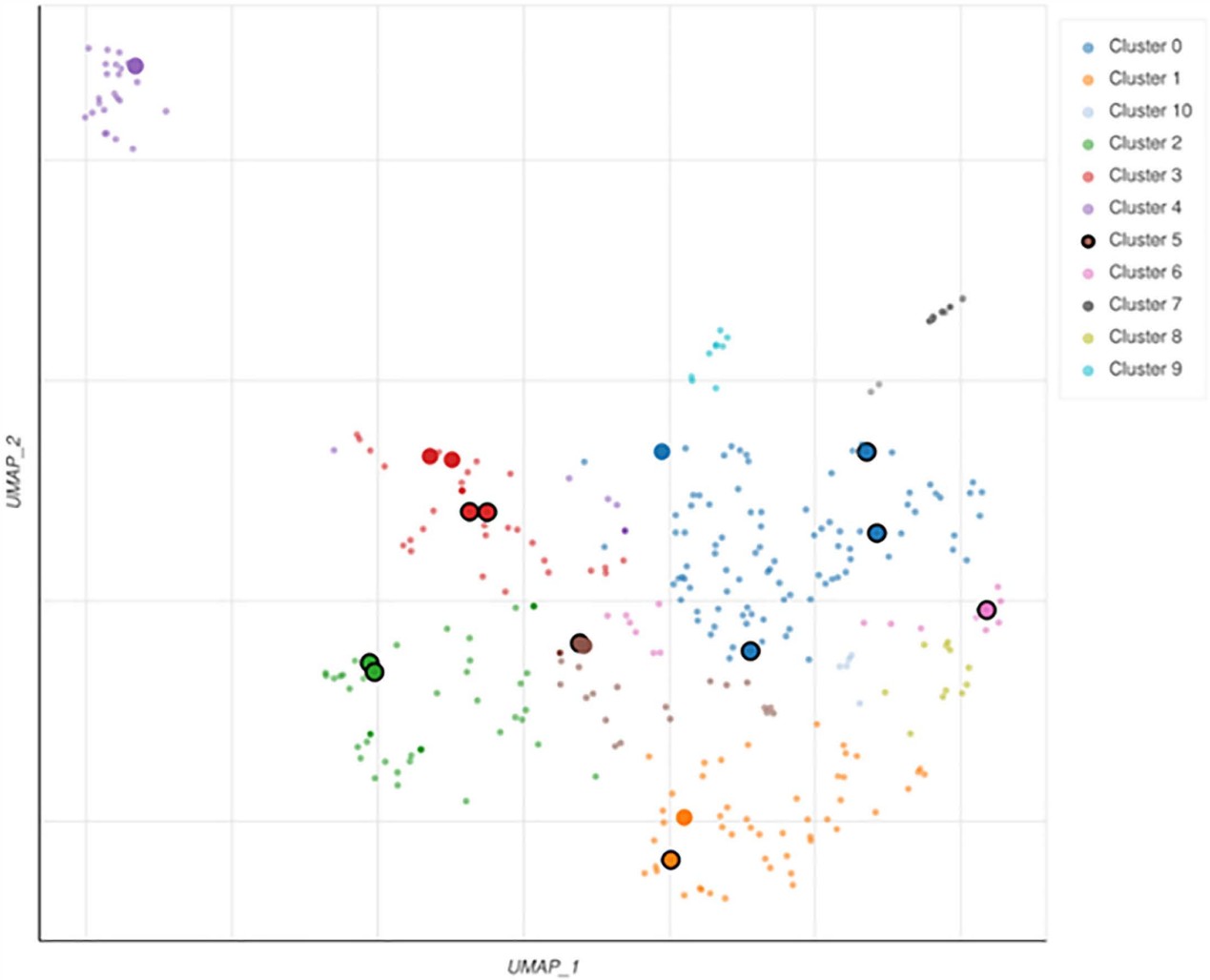

**Fig 11. Leiden algorithm-identified clusters for uniform manifold approximation and projection (UMAP) visualization of optimized gene sets.**
Each point represents a clustered set of genes. Large black points are substantially enhanced phrases that indicate dataset-related biological systems.

participating in various pathways including chylomicron clearance, HDL remodeling, and plasma lipoprotein assembly. The presence of CLDN5 in tight junction expression under RunX1 underlines its role in blood-brain barrier integrity, a key factor in vascular dementia. The enrichment of nuclear receptor signaling (APOE and MMP9) links lipid metabolism, inflammation, and endothelial function, confirming the multidimensional nature of the etiology of vascular dementia. These findings shed light on the molecular mechanisms underlying vascular dementia. Knowing these pathways helps uncover therapeutic targets and biomarkers for early vascular dementia diagnosis and intervention.

This volcano plot (Fig 13) provides a summary of the odds ratio (effect magnitude) and importance of the increased Reactome 2022 routes in the dataset. The x-axis (odds ratio) displays the degree of association between a route and the gene set under investigation, whereas the y-axis (-log(p-value)) indicates statistical significance. We are particularly interested in pathways with high odds ratios and significant importance (upper right quadrant) because these are the most physiologically relevant processes in vascular dementia. Paths with high statistical significance ($p < 0.05$) are highlighted by larger, darker blue spots, suggesting that they may play a role in disease processes. Numerous of these pathways are

**Table of top 10 significant p-values and q-values for Reactome Pathways 2024**

| term | p-value | q-value | overlap_genes |
|---|---|---|---|
| Metabolism of Vitamins and Cofactors | 0.001425 | 0.031250 | [MTHFR, APOE] |
| RUNX1 Regulates Expression of Components of Tight Junctions | 0.001499 | 0.031250 | [CLDN5] |
| Chylomicron Clearance | 0.001499 | 0.031250 | [APOE] |
| Signaling by Nuclear Receptors | 0.002628 | 0.031250 | [APOE, MMP9] |
| Chylomicron Assembly | 0.002997 | 0.031250 | [APOE] |
| Chylomicron Remodeling | 0.002997 | 0.031250 | [APOE] |
| HDL Remodeling | 0.002997 | 0.031250 | [APOE] |
| Metabolism of Folate and Pterines | 0.005090 | 0.040333 | [MTHFR] |
| Plasma Lipoprotein Assembly | 0.005687 | 0.040333 | [APOE] |
| Scavenging by Class A Receptors | 0.005687 | 0.040333 | [APOE] |

**Fig 12. The dataset streamlined the top ten main Reactome paths.** This table illustrates the paths with the most noteworthy p-values and q-values, underscoring their importance in the context of vascular dementia. Key pathways include nuclear receptor signaling, cofactor metabolism (chylomicron and HDL remodeling), and vitamin metabolism, with significant genes such as APOE, MTHFR, CLDN5, and MMP9.

most likely connected to lipid metabolism, vascular integrity, neuroinflammation, and endothelial function, which is in line with previous findings from studies of vascular dementia. Conversely, grey points indicate fewer statistically supported pathways that may not be directly associated with the disease or require further confirmation.

These plots identify the most relevant biochemical pathways associated with vascular dementia by providing a comprehensive overview of the Reactome Pathways 2024 enrichment study. In (Fig 14), the Manhattan plot (left) illustrates the statistical significance of the enriched pathways by showing paths with higher correlations at higher locations on the y-axis. Prioritizing the most crucial biological processes is facilitated by this graph, which highlights the significant terms with high enrichment. Paths with similar gene compositions are positioned closer to one another in (Fig 14) the UMAP scatterplot (Fig 14, right), which offers a functional clustering of gene sets. These phrases were grouped using the Leiden algorithm to enhance the visibility of functionally linked pathways. More significant upgraded pathways are highlighted by larger and darker points, which highlight their significance in the dataset. The distribution of terms suggests that the pathophysiology of vascular dementia may involve several biological processes, such as oxidative stress, lipid metabolism, and endothelial function. When combined, these investigations enhance our understanding of the fundamental biological underpinnings of vascular dementia, enabling targeted exploration of the most enriched pathways for potential treatment or biomarker development.

### 5) Protein sequence retrieval and modelling/Ligand retrieval

The protein sequences were retrieved from UniProt, 3D modelling was performed via trRosetta, and structural validation was performed for each protein via a Ramachandran analysis plot and Errat Score analysis, which indicated a stable configuration with most of the residues lying in the acceptable region. Furthermore, four natural ligands were selected after literature analysis, and their structures were retrieved from PubChem. The pharmacological profiles were evaluated for the ligands and are summarized in Table 3.

To examine the chemical diversity of the selected natural chemicals (curcumin, galangin, resveratrol, and licocumarone), we calculated pairwise Tanimoto similarity scores using Morgan fingerprints. All pairwise similarity values were less than 0.3, indicating that the chemicals are structurally diverse and form different chemical scaffolds. This structural

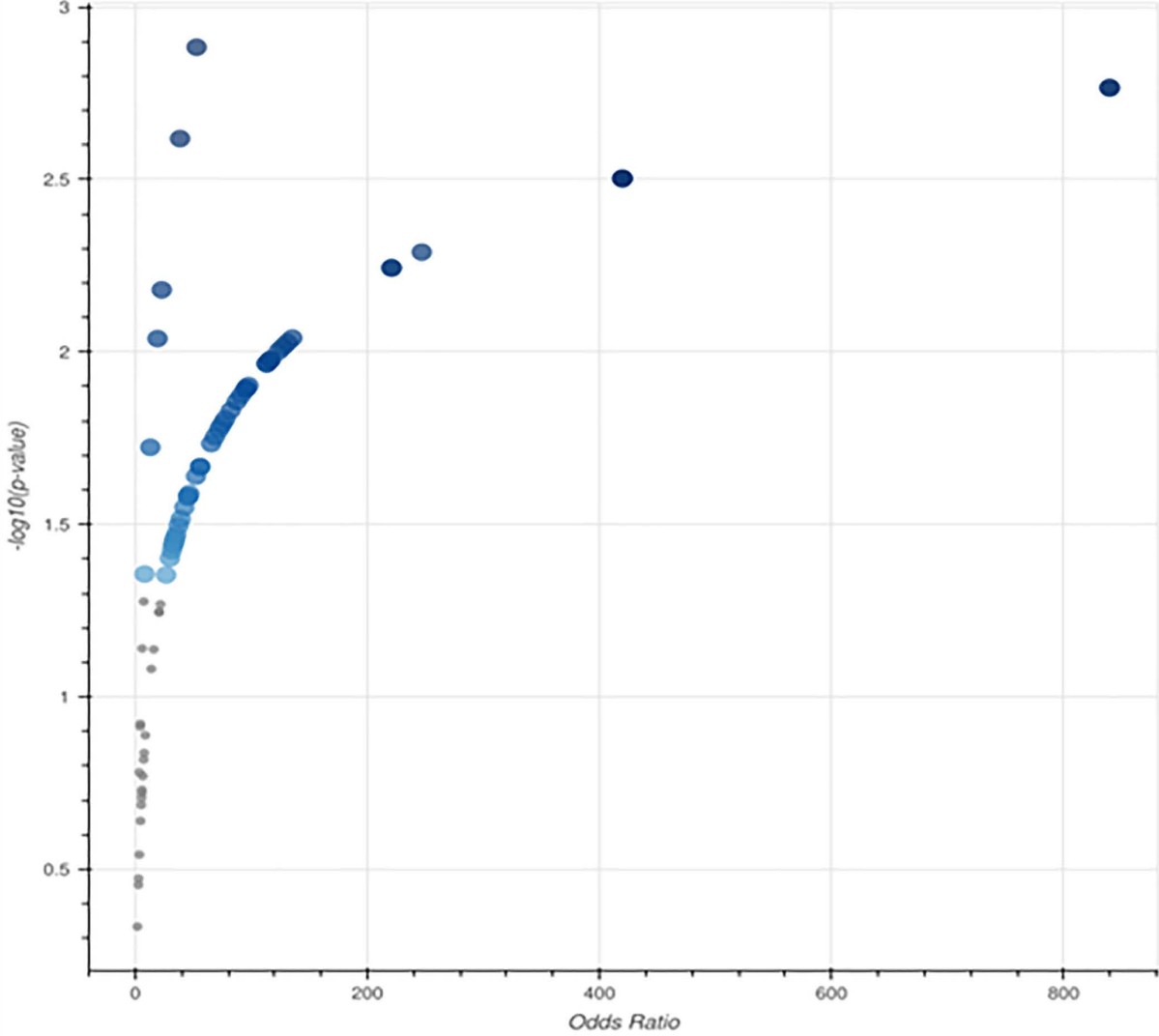

**Fig 13. Volcano plot illustrating the significance and odds ratio of gene sets from the Reactome 2022 pathway study.** The odds ratio is represented on the x-axis, and the -log(p-value) for each gene set is depicted on the y-axis. Larger blue points indicate significantly increased routes (p-value < 0.05), with darker shades representing greater relevance, and grey points indicate non-significant terms.

heterogeneity promotes their ability to influence many molecular targets associated with vascular dementia. The detailed similarity matrix is shown in Table 4. We selected Curcumin, Galangin, and Resveratrol for their anti-inflammatory properties in vascular dementia treatment. While Licocumarone has not been previously investigated in the context of neurodegenerative diseases, its coumarin derivatives have shown vascular and neuroprotective relevance in related compounds. Therefore, it was selected as the novel compound for this study.

The ADMET properties of four natural ligands curcumin, galangin, resveratrol, and licocumarone, were compared to two licensed CNS drugs; donepezil and memantine. Table 5 shows that the four natural substances have comparable blood-brain barrier permeability (logBB ~ 0.02–0.12), which is regarded acceptable for CNS medication candidates. All compounds showed good gastrointestinal (GI) absorption, no mutagenicity (negative AMES test), and no projected hepatotoxicity, which mirrored the safety profiles of the reference medications.

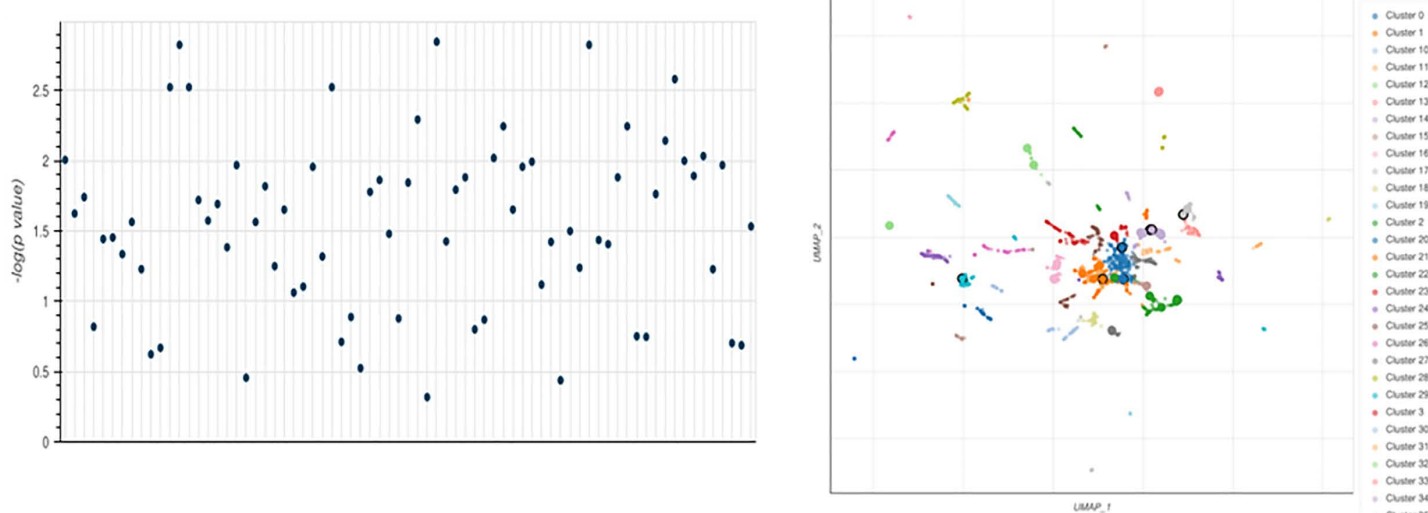

**Fig 14. (Left) Manhattan plot showing the significance of phrases enhanced by Reactome Pathways 2024.** Every point denotes a unique path, and the and y-values (-log₁₀(p-value)) show statistical significance. (Right) UMAP-based clustering of enriched phrases, where similar paths are arranged closer together. Larger and darker marks indicate more statistically significant terms. The Leiden method was used for clustering.

**Table 3. Comparative pharmacological profile of selected natural compounds evaluated in this study.**

| Compound | Natural Source | Bioavailability | Neuroprotective Mechanism | Evidence in Neurodegeneration | Key References |
|---|---|---|---|---|---|
| **Curcumin** | *Curcuma longa* (Turmeric) | Low; improved with piperine or nanoparticle carriers | Antioxidant, anti-inflammatory, inhibits amyloid aggregation | Shown to reduce Aβ burden and rescue neuronal damage in AD and VaD models | Maiti and Dunbar (9) |
| **Galangin** | *Alpinia officinarum*, honey | Moderate | Free radical scavenger; inhibits NF-κB and iNOS pathways | Reduces neuroinflammation and ROS in various experimental models | Kabel, Arab [34] |
| **Licocumarone** | Licorice root derivatives | potential coumarin-like behavior | Antioxidant and anti-inflammatory effects through coumarin scaffold | Novel compound tested on the; evidence based on in silico affinity and structural similarity to known neuroprotective coumarins | Ibrahim, Skalicka-Woźniak [35] |
| **Resveratrol** | Grapes, red wine | Low to moderate; enhanced with liposomal delivery | Activates SIRT1, reduces oxidative stress and Aβ accumulation | Shown to improve cognitive outcomes in AD; enhances cerebral blood flow | Wang, Wang [36] |

**Table 4. Tanimoto similarity matrix displaying pairwise structural similarity among chosen natural ligands based on Morgan fingerprints.**

| Compound | Curcumin | Galangin | Resveratrol | Licocumarone |
|---|---|---|---|---|
| **Curcumin** | 1.0 | 0.1 | 0.27 | 0.18 |
| **Galangin** | 0.1 | 1.0 | 0.13 | 0.16 |
| **Resveratrol** | 0.27 | 0.13 | 1.0 | 0.11 |
| **Licocumarone** | 0.18 | 0.16 | 0.11 | 1.0 |

**Table 5. Comparison of predicted ADMET properties of natural compounds and approved CNS drugs.**

| Property | Donepezil | Memantine | Curcumin | Galangin | Resveratrol | Licocumarone |
|---|---|---|---|---|---|---|
| Blood-Brain Barrier (logBB) | 0.09 | 0.31 | ~0.02 | ~0.10 | ~0.08 | ~0.12 |
| GI Absorption | High | High | Moderate | High | High | High |
| CYP450 Inhibition | CYP2D6 inhibitor | No | CYP1A2, 2C9 inhibitor | CYP1A2 inhibitor | CYP1A2, 3A4 inhibitor | No |
| Total Polar Surface Area (TPSA) | 38.77 Å² | 24.39 Å² | 93.06 Å² | 90.90 Å² | 60.69 Å² | 66.76 Å² |
| AMES Toxicity | No | No | No | No | No | No |
| Hepatotoxicity | No | No | Low | No | No | No |

Additionally, their total polar surface area (TPSA) values (60–93 Å²) are within the range for successful CNS penetration (usually <90 Å²), with Galangin and Resveratrol exhibiting particularly excellent values. The CYP450 inhibition patterns varied significantly but remained within drug-tolerable limits, with no pan-inhibition observed. These characteristics confirm that the chosen natural ligands not only show promising biological activity, but also meet several important pharmacokinetic benchmarks for CNS drug development.

## 6) Molecular docking analysis

Molecular Docking performed using AutoDock Vina showed different results for each compound Table 6; the active sites were identified using Protein Plus, and docking was performed. Furthermore, visualization was performed using Chimera, and the interactions were analyzed in detail using LigPlot. Protein Amyloid beta precursor-like protein 1 showed the best interactions with galangin, by three hydrogen bonds and several hydrophobic interactions (Fig 15).

Moreover, docking was performed for all four ligands with another targeted protein, Apolipoprotein E, Curcumin showed the best results with 3 hydrogen bonds (Fig 16).

Similarly, Claudin-5 showed the best interaction with licocumarone (Fig 17), indicating two hydrogen bonds. Matrix metalloproteinase-9 had the best interaction with Curcumin, with 3 hydrogen bonds. Similarly, Methylene tetrahydrofolate showed optimum interaction with resveratrol, with four hydrogen bonds. Furthermore, Superoxide dismutase showed bonding with licocumarone and four hydrogen bonds, as shown in Figs 18–20, respectively.

Docking data showed that binding energies were consistent across all five runs for each complex. The standard deviations varied from 0.42 to 1.07 kcal/mol, demonstrating limited variance and validating the predictability of the binding modes. For example, APLP1-Galangin had the lowest mean score (−6.80 kcal/mol) and an SD of 1.07, whereas SOD1-Licocumarone had the most stable interaction (SD = 0.42). These findings support the reliability of docking and the strength of the chosen binding conformations (see Supplementary Table 1 and 2 in S1 File for full data).

**Table 6. Summarizes the overall number of interactions and their bonding energies.**

| Proteins | Ligands | Docking Score | Number of interactions (H-bonds) |
|---|---|---|---|
| Amyloid beta precursor-like protein 1 | Galangin | −8.5 kcal/mol | 3 |
| Claudin-5 | Licocumarone | −6.8 kcal/mol | 2 |
| Matrix metalloproteinase-9 | Curcumin | −8.0 kcal/mol | 3 |
| Methylenetetrahydrofolate | Resveratrol | −8.1 kcal/mol | 3 |
| Superoxide dismutase | Licocumarone | −6.6 kcal/mol | 4 |
| Apolipoprotein E | Curcumin | −7.2 kcal/mol | 2 |

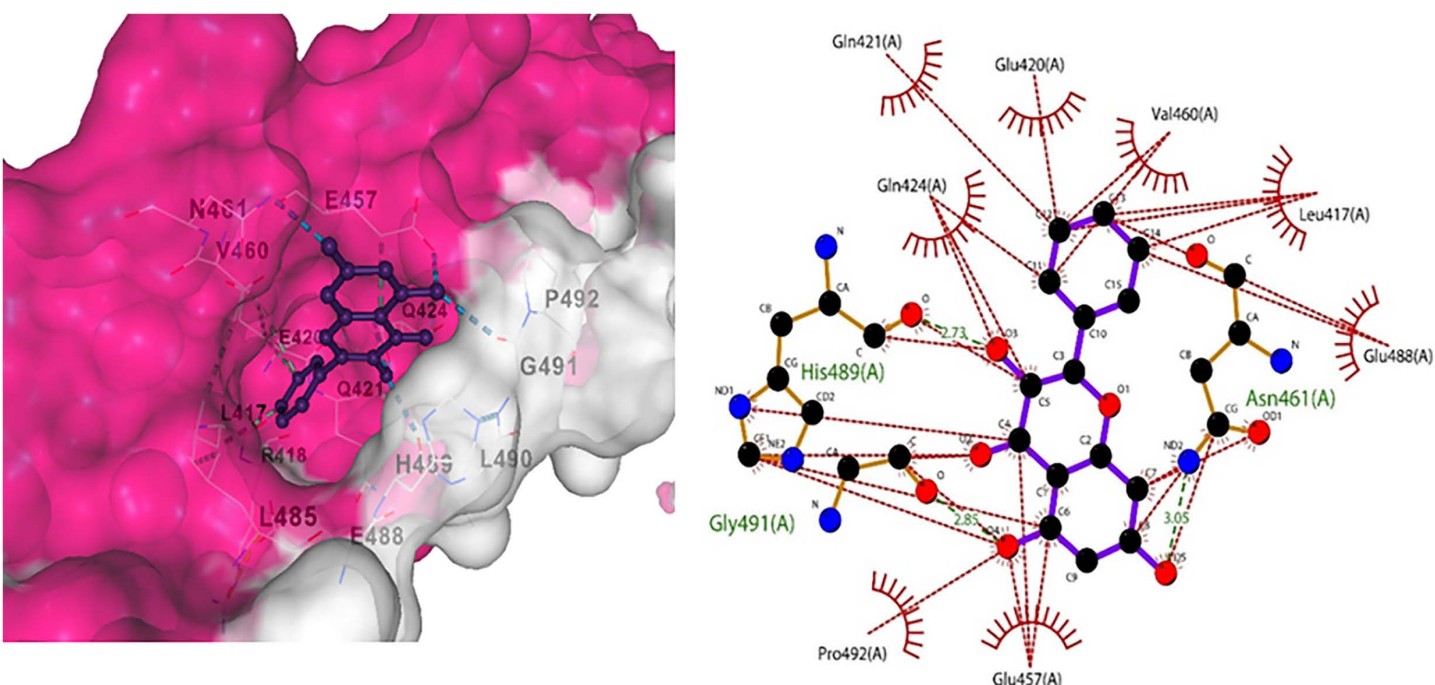

**Fig 15. (a) Docking interaction for Galangin with Amyloid protein, (b) Indicates hydrogen bond in green and hydrophobic interactions in red.**

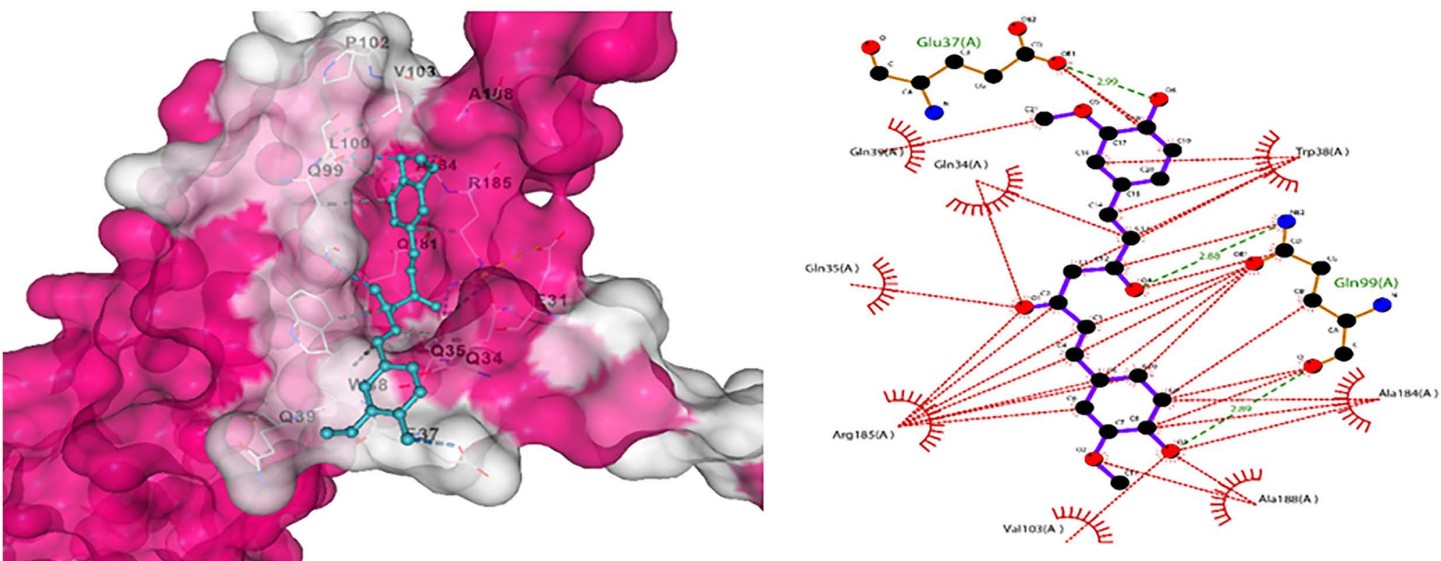

**Fig 16. (a) Indicates docking interaction for Apolipoprotein with Curcumin. (b) Interaction analysis for docked complex presenting green colour for hydrogen bond and red for hydrophobic interactions.**

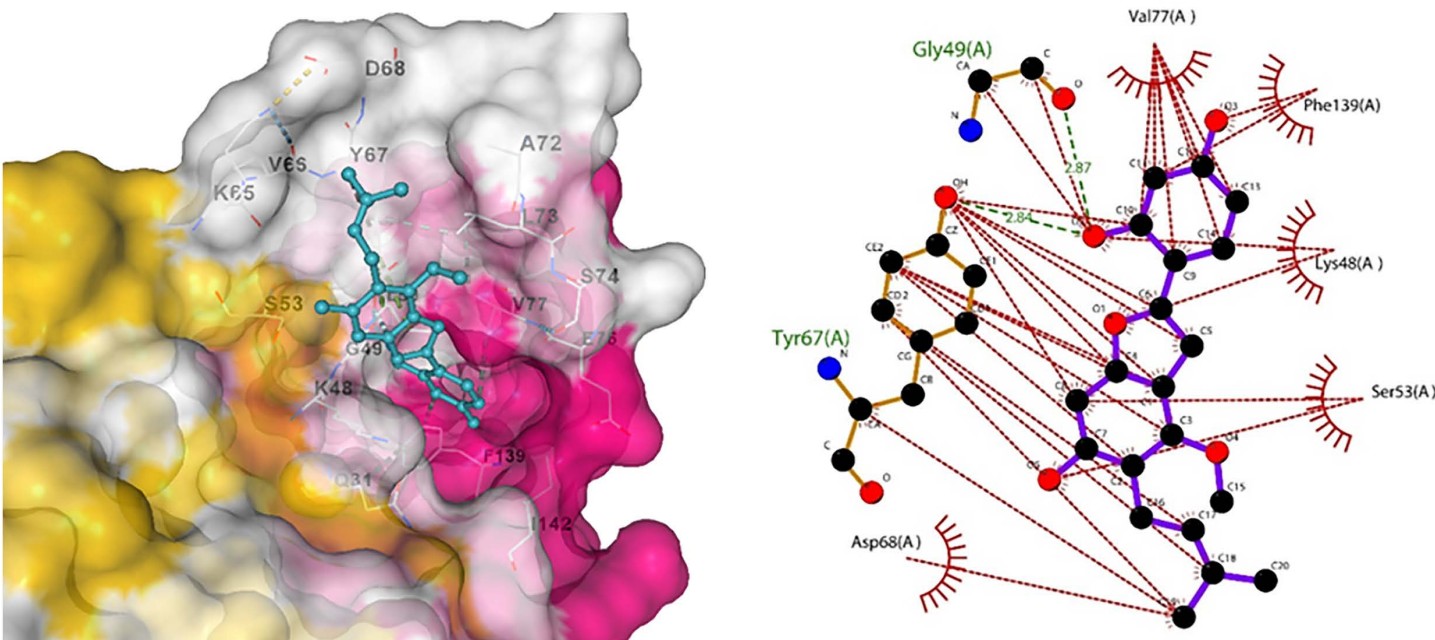

**Fig 17. (a) Indicated bonding analysis of Claudin-5 with Licocumarone, (b) indicates the docked complex's hydrogen bonds and other hydrophobic interactions.**

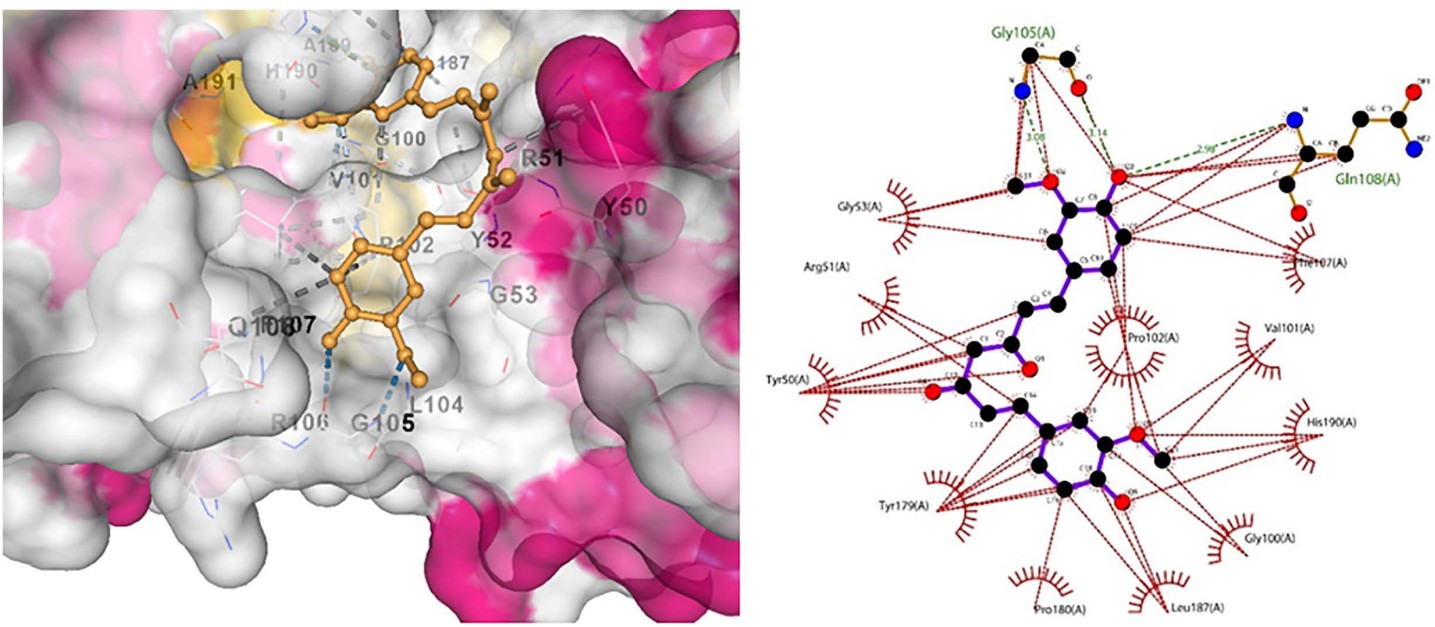

**Fig 18. (a) Indicated bonding interaction of Curcumin with Matrix metalloproteinase-9 (b) elaborates the hydrogen bonds in green and the hydrophobic interaction in red.**

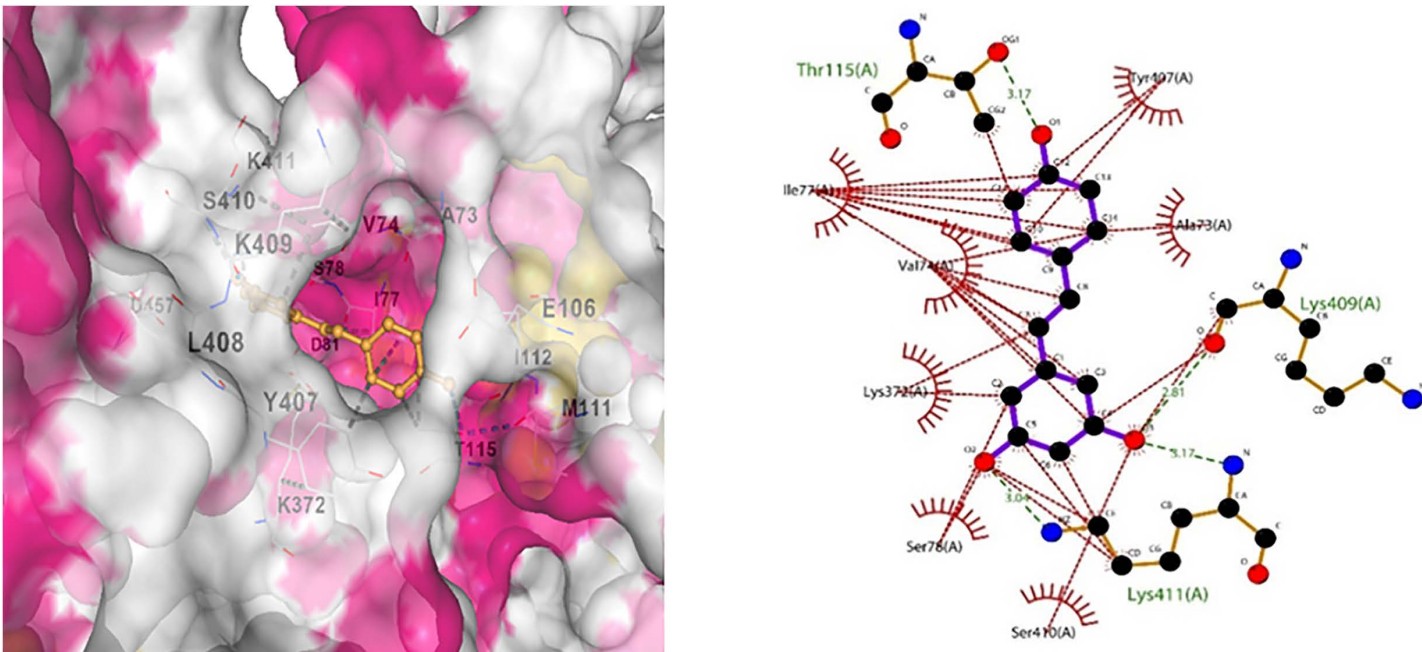

**Fig 19. (a) Indicated bonding interaction of Resveratrol with Methylene tetrahydrofolate. (b) Elaborates the hydrogen bonds in green and the hydrophobic interaction in red.**

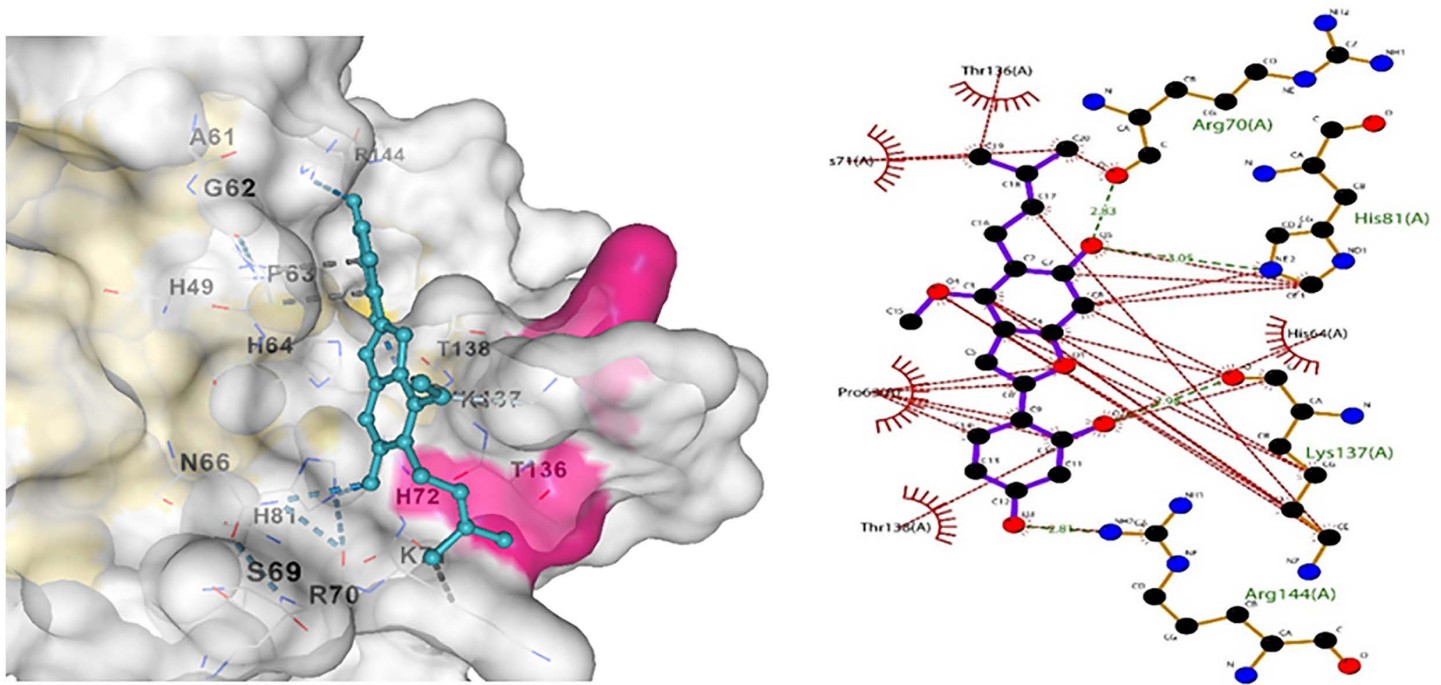

**Fig 20. (a) Indicated bonding interaction of Licocoumarone with Superoxide dismutase. (b) Elaborates the hydrogen bonds in green and the hydrophobic interaction in red.**

## 7) Normal mode analysis

iMODs utilize Normal Mode Analysis (NMA) to assess the molecular mobility and structural flexibility of the linked complexes. The principle of NMA of proteins posits that the "soft modes," characterized by the lowest-frequency vibrational normal modes, represent the most substantial motions of the protein and are therefore functionally relevant. The analysis of biomolecular dynamics has been significantly facilitated by normal-mode analysis, which is a reliable and advantageous method. Moreover, it is exceptionally efficient to examine protein processes that transpire incrementally and undergo significant conformational alterations (Skjaerven et al., 2009; Soltan et al., 2021).

The Normal Mode Analysis (NMA) results for APLP1 mixed with galangin are shown in this study, along with information on the structural flexibility and how it changes over time. NMA studies on the intrinsic motions of proteins have shown flexible patches, collective motions, and possible functional locations. In (Fig 21a), the B-factor graph shows how NMA predicted residue flexibility (in red) compared to PDB's actual B-factor values (in black). A higher atomic variation indicates a larger B-factor. It is clear from this method that the loops and terminal parts are adjustable, whereas the basic structural parts are not adjustable. The computer method overstates changes because it employs an elastic network model; however, the NMA predictions align with the actual B-factors. The deformability line in (Fig 21b) shows the parts of proteins that can change their structure. Peaks in this graph show residues that are very mobile and are found in loops or close to sites where ligands bind. The flexible parts of APLP1 may be important for binding galangin or for changing its shape.

The variance contribution plot in (Fig 21c) shows the extent to which the first 20 normal modes affect protein motion. The first few modes indicate that APLP1's overall movements can be summarised as a few main modes that account for most of the variations. There is more variance as the number of modes increases; however, the key structural motions

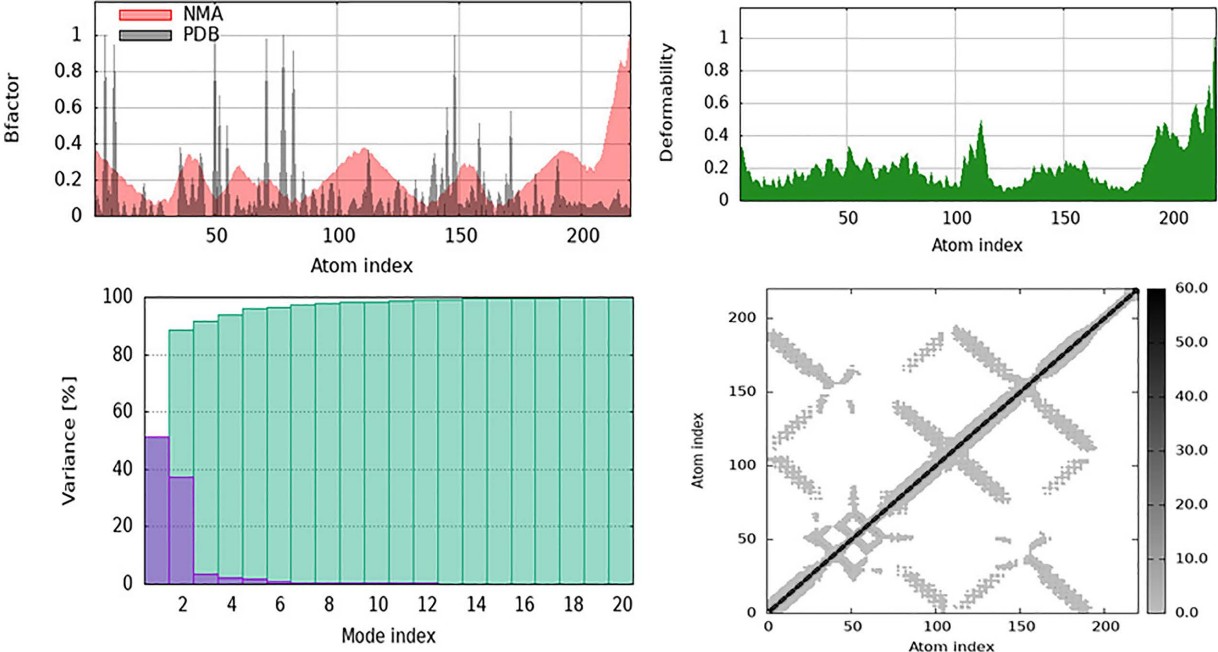

**Fig 21. The Normal Mode Analysis (NMA) results for amyloid beta precursor-like protein 1 (APLP1) complexed with Galangin show that the structure is flexible and changes over time.** (a) A B-factor graph that shows areas that are flexible and stiff, comparing the NMA-predicted flexibility (red) with the experimental PDB B-factor (black). The deformability graph shows the mobility of specific residues, with peaks showing the most flexible places. The variance input of the first 20 normal modes shows that there aren't many dominant modes that can fully describe the motion of the structure. (d) A cross-correlation grid that shows correlated (dark areas) and anti-correlated (lighter areas) atomic movements shows how the protein's cooperative domains move.

occur in the first 10 modes. This implies that APLP1's basic movements are probably mostly managed by low-frequency motions. In (Fig 21d), the cross-correlation matrix shows the relationship between the movements of protein domains. As expected, the black diagonal line shows self-correlations, whereas elements that are not on the diagonal show residue pairs that are linked or anticorrelated. Strong connections indicate coordinated domain movements, which can alter how proteins work or interact with ligands. APLP1 has different correlation patterns, suggesting that some parts move together, potentially controlling allosteric responses or structural reorganization when galangin binds.

The significant flexibility seen in the loop regions surrounding the E2 domain suggests that Galangin binding could use induced fit mechanisms, stabilize amyloidogenic surface grooves. The core domain stiffness promotes stable ligand anchoring, whereas loop flexibility may influence APP cleavage or oligomerization behavior relevant to VaD. Normal Mode Analysis (NMA) demonstrated that curcumin binding influences the structural flexibility and dynamic behavior of ApoE by stabilizing its core and altering localized flexibility. The B-factor plot (a) in (Fig 22) indicates that the core structure exhibits fewer variations, even though the termini and loop areas remain malleable. This suggests that curcumin enhances the structural stiffness in functionally relevant domains. The deformability plot in (Fig 22b) indicates that curcumin may significantly influence residues involved in binding interactions or conformational changes by altering their flexibility in specific regions. Although the fundamental dynamics of ApoE are still regulated by a few dominant low-frequency modes, the variance plot in (Fig 22c) suggests that there are minor changes in the variance distribution, suggesting localized structural alterations as a result of ligand binding. Additionally, the cross-correlation matrix (Fig 22d) indicated that curcumin affected the correlated and anti-correlated motions between various regions of the protein, thereby influencing its allosteric control and functional relationships. The binding of curcumin appears to stabilize ApoE in general, while modifying its flexibility

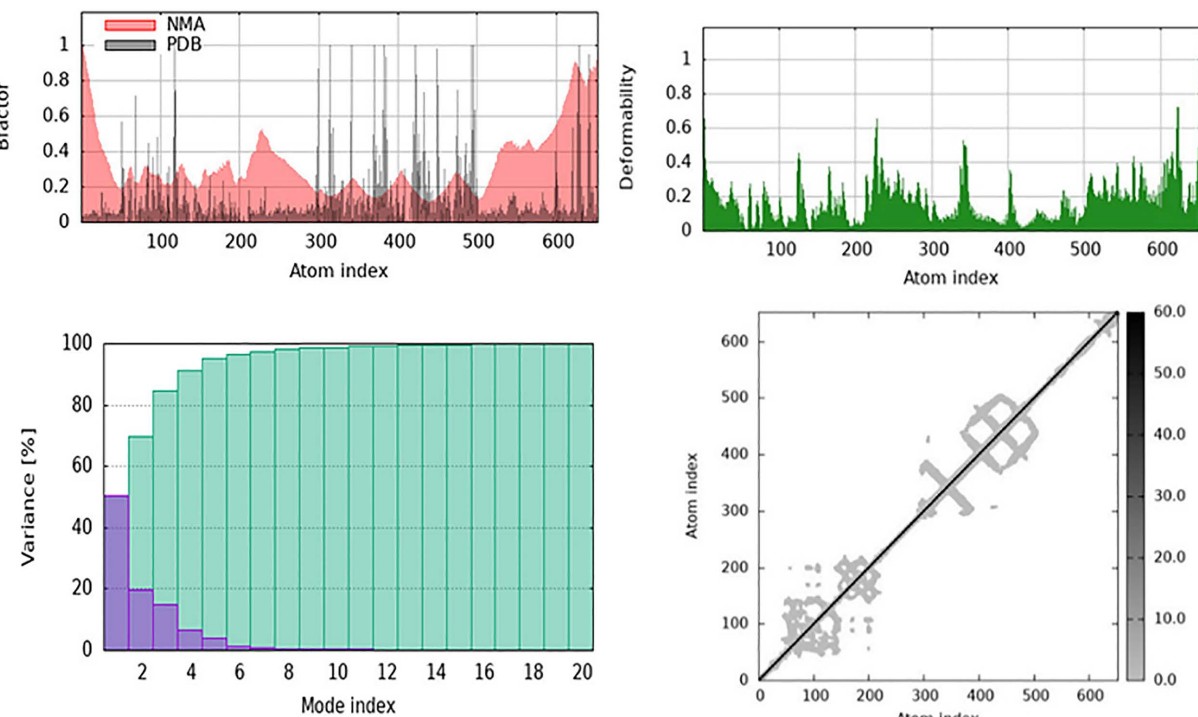

**Fig 22. Normal Mode Analysis for Apolipoprotein E the structural dynamics of the curcumin-complexed Apolipoprotein E (ApoE) are displayed.** Curcumin binding stabilizes the core while controlling flexibility in certain regions, as seen in the B-factor and deformability graphs. Variance distribution and cross-correlation analysis suggest altered cooperative motions and localized structural alterations.

in specific regions. This effect may affect ApoE's function in lipid transport, receptor interactions, and neurodegenerative disorders.

Curcumin binding increased flexibility near APOE's receptor-binding region, potentially impacting its interactions with lipoprotein receptors and β-amyloid clearance machinery. Stabilization of the core domain implies structural integrity, although changed associated movements indicate a potential impact on lipid transport and allosteric regulation. Furthermore, the NMA data for Claudin-5 with Licocumarone indicate that ligand binding affects the cooperative movements, flexibility, and stability of the protein. Licocumarone stabilized the core transmembrane regions while permitting increased flexibility in the loop and terminal areas, potentially facilitating the conformational changes required for function, as indicated by the B-factor (Fig 23a). The deformability shown in (Fig 23b) indicates specific regions with altered flexibility, suggesting that ligand binding influences dynamic residues involved in permeability regulation. The primary motions of Claudin-5 were concentrated in a limited number of significant modes, as indicated by the variance plot in (Fig 23c). This suggests that the ligand does not substantially affect global dynamics but induces localized alterations. Changes in cooperative domain motions indicated by the cross-correlation matrix (Fig 23d) suggest that licocumarone alters inter-residue interactions, potentially affecting the role of Claudin-5 in tight junction stability and blood-brain barrier function.

Flexibility in extracellular loops, as shown in the deformability figure, may reflect tight junction strand conformational flexibility, which influences BBB permeability. Licocumarone's capacity to maintain transmembrane domains while permitting loop mobility may improve barrier modulation during inflammatory stress, which is a hallmark of VaD. Moreover, the NMA results for Matrix Metalloproteinase-9 (MMP-9) indicate the structural and dynamic effects of curcumin on ligand binding. Curcumin stabilizes the protein, reduces fluctuations in most places, and maintains termini flexibility Fig 24, B-factor plot, a). The deformability map (Fig 24 b), highlights the flexible sections at the N- and C-terminals that are required for localized motion, as suggested by enzyme activity and substrate interactions. The variance plot in (Fig 24c) shows that the first few normal modes dominate the protein dynamics, suggesting that curcumin slightly alters the

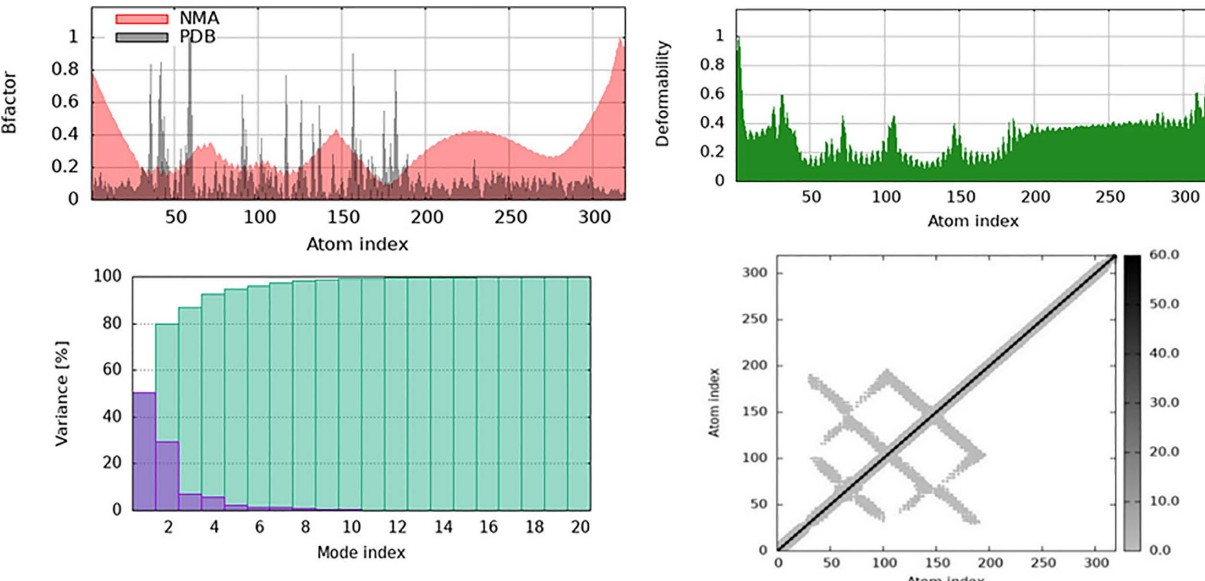

**Fig 23. NMA data for Claudi-5 complexed with Licocumarone showing structural dynamics.** As shown in B-factor and deformability graphs, ligand binding stabilises the transmembrane core and increases loop and terminal flexibility. Cross-correlation analysis and variance distribution suggest localised conformational changes may alter Claudin-5's tight junction stability and permeability.

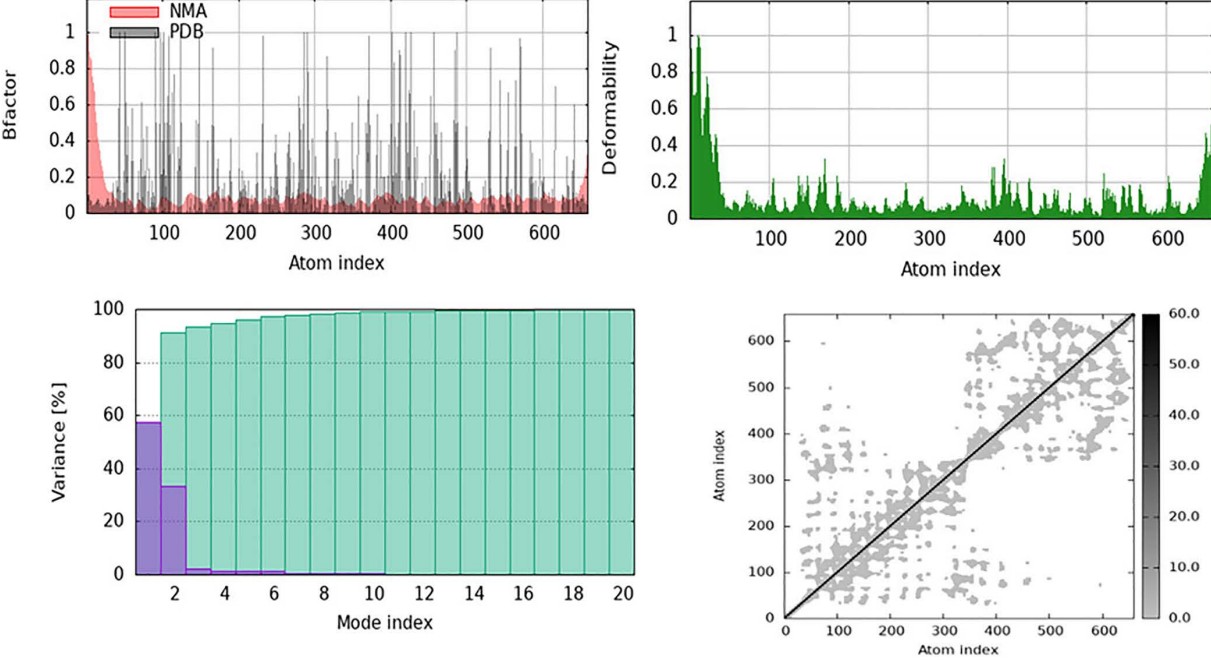

**Fig 24. Normal Mode Analysis Matrix metalloproteinase-9.** Structural dynamics of Matrix Metalloproteinase-9 (MMP-9) complexed with curcumin are shown by normal mode analysis (NMA) data. As seen in B-factor and deformability graphs, curcumin binding stabilises the protein core while preserving flexibility at terminal areas. Localised conformational alterations shown by variance and cross-correlation analysis could affect substrate interactions and enzymatic activity of MMP-9.

conformation. Cross-correlation matrix (Fig 24d) shows variations in correlated motion over distant locations, suggesting that ligand binding affects cooperative domain contact, MMP-9 catalytic efficiency, and substrate recognition.

Curcumin may modulate substrate access and enzymatic activity via deformable loop regions in the catalytic cleft. Stabilization of the surrounding domains may help to selectively block ECM-degrading function, lowering BBB disruption and neuroinflammation in VaD. Furthermore, the NMA results for Methylenetetrahydrofolate Reductase (MTHFR) with resveratrol revealed that ligand binding affects protein structural stability and flexibility. Resveratrol stabilizes MTHFR's major regions of MTHFR while allowing some flexibility at the terminal ends, according to the B-factor plot (Fig 25a), which is important for its enzymatic function. The deformability plot in (Fig 25b) reveals that resveratrol binding produces localized conformational changes that may affect allosteric regulation or substrate interactions. Variance plot (Fig 25c) shows that the first few normal modes of the protein dominate, indicating that resveratrol binding fine-tunes essential mobility rather than disrupting structural dynamics. Resveratrol affects long-range correlated movements, which may alter MTHFR's enzymatic efficiency and functional regulation of MTHFR, as observed in the cross-correlation matrix (Fig 25d).

Local flexibility in terminal sections and loop segments near the cofactor binding site may enable substrate exchange or allosteric regulation. Resveratrol may thereby stabilize the enzyme while increasing its metabolic throughput, potentially reducing hyperhomocysteinemia-induced vascular damage. Moreover, the results of NMA for Superoxide Dismutase (SOD1) with licocumarone indicate that ligand binding induces structural flexibility while maintaining core stability. Licocumarone stabilizes certain regions while increasing loop and terminal segment fluctuations, as seen in B-factor (plot) (Fig 26a). This could have an impact on the binding interactions or enzymatic activity. The deformability plot in (Fig 26b) indicates significant flexibility throughout the protein, suggesting that ligand interaction does not induce rigidity but allows for dynamic conformational changes. The initial normal modes notably contribute to the overall motion, as indicated by

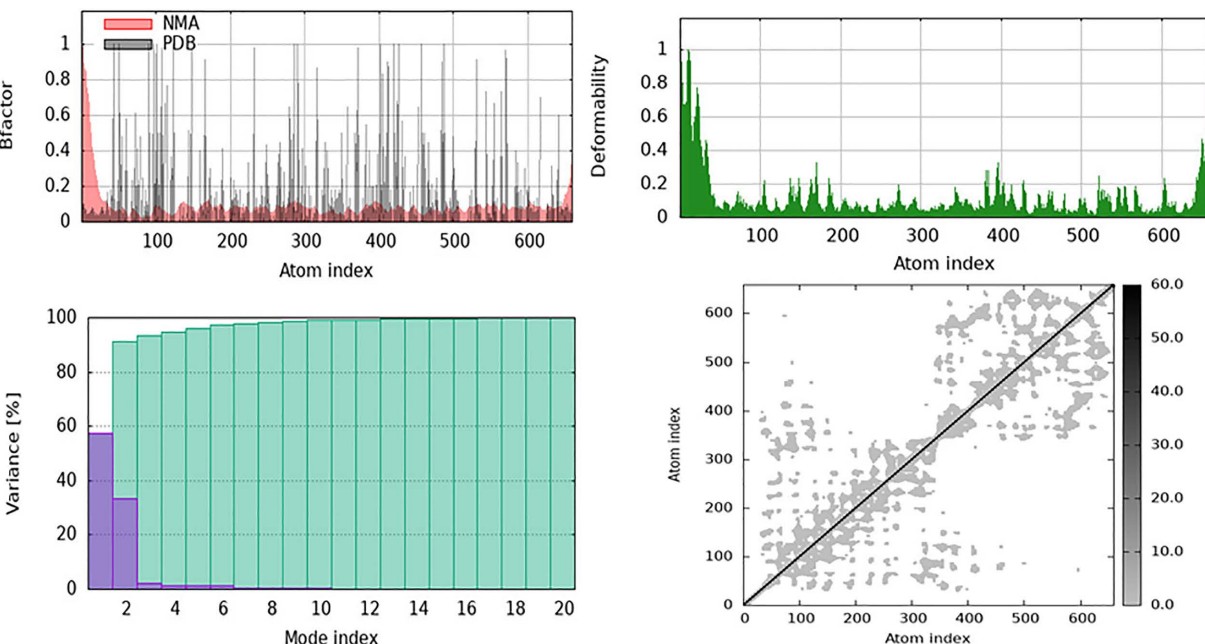

**Fig 25. Methylenetetrahydrofolate Reductase (MTHFR) complexed with Resveratrol shows structural dynamics under Normal Mode Analysis (NMA).** B-factor and deformability graphs show that resveratrol binding stabilizes the core and maintains terminal flexibility. Variance and cross-correlation studies demonstrate local conformational changes that may affect MTHFR's enzymatic activity and functional regulation.

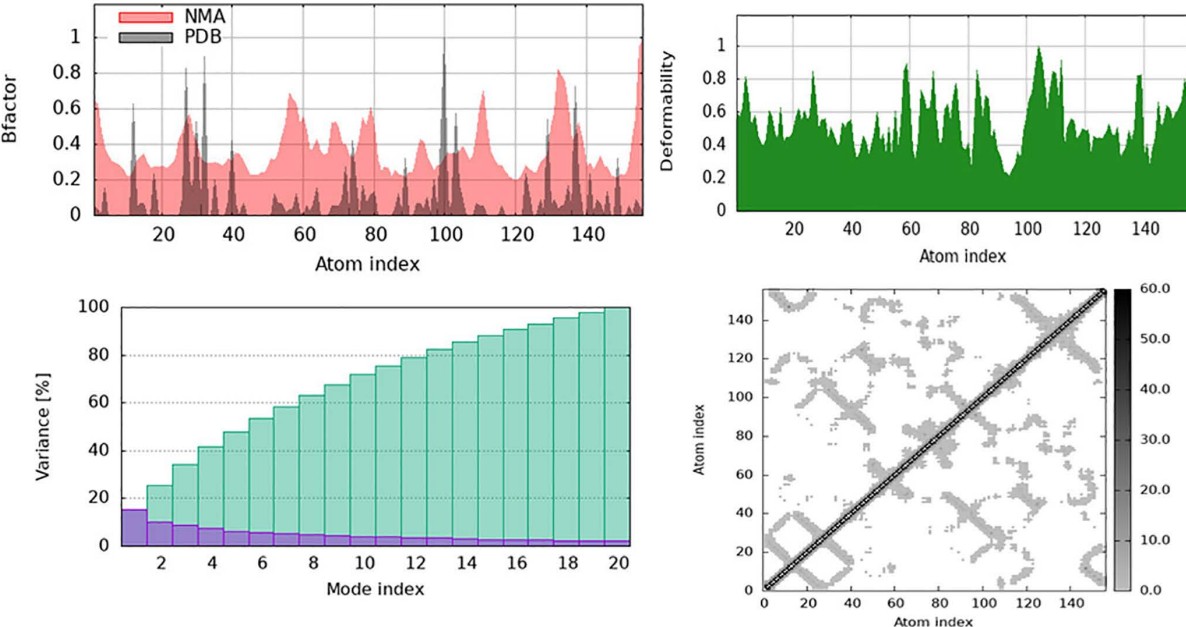

**Fig 26. Normal Mode Analysis of Superoxide dismutase.** The NMA data for SOD1 complexed with licocumarone show its structural dynamics. B-factor and deformability graphs show that licocumarone binding increases loop and terminal flexibility while retaining core stability. Variance and cross-correlation data suggest that ligand-induced conformational changes may affect SOD1 functional interactions and enzymatic activity.

the variance plot in (Fig 26c), which suggests that licocumarone binding affects large-scale conformational changes while maintaining global stability. The cross-correlation matrix (Fig 26d) indicates altered cooperative motions between distant regions, suggesting that licocumarone influences long-range interactions within SOD1, potentially impacting its catalytic activity and stability. Therefore, flexibility in loop and surface-exposed β-strands may affect metal ion coordination and ROS scavenging. Stabilization of SOD1's core suggests that Licocumarone may preserve enzymatic function while increasing structural resilience to oxidative stress, thereby contributing to neuroprotection.

## MD simulation analysis

We used RMSD, RMSF, radius of gyration (Rg), and solvent-accessible surface area (SASA) analysis to evaluate the dynamic behaviour and structural stability of protein-ligand complexes across a 100 ns simulation period. The RMSD profiles (Fig 27a) showed early convergence (~30 ns) for all systems, with Amyloid beta precursor-like protein 1 – Galangin and Matrix metalloproteinase-9 – Curcumin having the lowest RMSD values (~0.30–0.32 nm), indicating stable interactions. Apolipoprotein E – Curcumin and Superoxide Dismutase – Licocumarone complexes showed slightly greater variations (~0.45–0.47 nm), indicating flexible binding. RMSF analysis (Fig 27b) indicated residue-specific changes compatible with structural mobility near flexible loops and termini. Resveratrol and Curcumin-bound compounds exhibited negligible per-residue variations (RMSF < 0.08 nm), indicating overall backbone stiffness. Complexes containing Licocumarone, on the other hand, showed greater RMSF peaks in loop areas, which could indicate localized flexibility. The Rg plot (Fig 27c) confirmed the observations, with Galangin and Curcumin systems maintaining lower Rg values (~1.65–1.70 nm), indicating compact, folded structures. Apolipoprotein E and Claudin-5 complexes exhibited modest fluctuation in Rg (~1.50–1.60 nm), suggesting transient unfolding processes.

The SASA results (Fig 27d) verified these dynamics. Stable binders like Galangin and Curcumin maintained consistent surface exposure (~96–97 nm²), while less stable complexes like Apolipoprotein E had reduced SASA (~92–93 nm²), possibly due to local conformational changes. These simulations support the docking results by demonstrating that Galangin, Curcumin, and Resveratrol form strong and structurally stable complexes with their respective targets.

Although each simulation was performed for 100 ns, all systems indicated early convergence (<30 ns) and steady trajectories thereafter, reducing the possibility of stochastic artifacts. To evaluate the stability and reproducibility of dynamic metrics (RMSD, RMSF, Rg, and SASA), time-windowed averaging was performed across the final 50 ns of each trajectory. Within this window, RMSD fluctuations were less than 0.06 nm, while the radius of gyration was stable within ±0.04 nm. It is supported by the comparatively low variance in structural metrics that the chosen ligands exhibited stable, compact complex formations with the target proteins. While replicate simulations were not carried out due to computational constraints, the consistency of time-resolved metrics and the early plateauing of RMSD profiles increase confidence in the observed conformational behavior.

## Discussion

Vascular Dementia is the most common form of dementia and is marked by a high mortality rate. Research shows that vascular dementia patients have a 3.33 mortality risk than Alzheimer's patients. A Chinese study found 28.91% of in-hospital deaths per 1,000 person-years [37]. Several factors affect vascular dementia, such as advanced age, male gender, concomitant disorders such as hypertension and cardiac disease, cognitive decline, and increased risk of death. Vascular dementia is severe, and patients have a lower life expectancy than those with other types of dementia; hence, managing risk factors may improve survival [38]. In this study, we performed Docking analysis with natural ligands to analyze the major proteins involved in different pathways associated with this deadly disease and to identify a convenient and effective therapeutic target that will prove more efficient in treating vascular dementia. We selected four natural ligands based on their oxidative stress, anti-neuroinflammatory, blood-brain barrier integrity, and amyloid deposition properties.

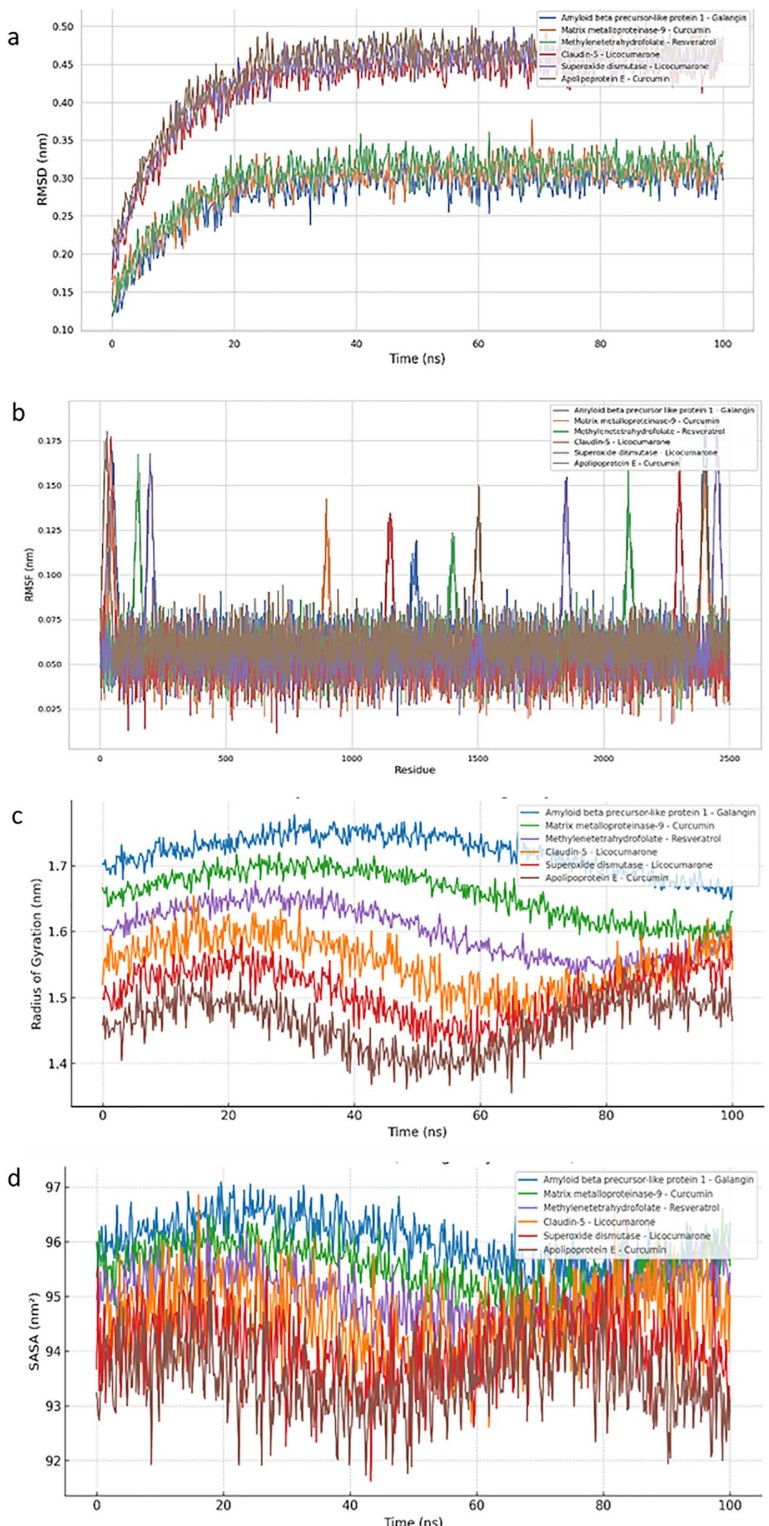

**Fig 27. Shows a molecular dynamics simulation of six protein-ligand complexes spanning 100 ns.** (a) RMSD profiles that demonstrate convergence and structural integrity. (b) The RMSF per-residue fluctuation analysis reveals flexible loop areas. (c) The radius of gyration profiles reflects compactness and overall stability. (d) SASA profiles showing surface exposure of protein-ligand complexes. Curcumin, galangin, and resveratrol compounds show greater structural stability across all criteria.

Furthermore, several analyses were conducted to confirm the significance of these proteins. The String database was utilized for protein-protein interaction (PPI) analysis, and functional enrichment analysis was performed, with a particular focus on the KEGG and Reactome databases to examine these proteins' functions in detail and their influence on vascular dementia. Furthermore, the cross-species conservation observed in the co-expression analysis highlights the evolutionary importance of the selected genes APOE, APLP1, MMP9, CLDN5, SOD1, and MTHFR, in controlling neuroinflammatory reactions and vascular function. Comparable expression patterns in humans and widely used model species like *Gasterosteus aculeatus*, *Danio rerio*, and *Mus musculus* imply that these genes are a component of fundamental regulatory networks that have been preserved throughout the evolution of vertebrates. Given this conservation, it is likely that human oxidative stress, neurodegeneration, and vascular dysfunction share many molecular mechanisms. As a result, these model species offer biologically relevant systems for studying the processes of vascular dementia (VaD). Furthermore, the fact that these co-expression patterns are conserved provides evidence for the concept that these genes play a meaningful functional role in disease rather than simply reflecting species-specific variability. Therefore, this evolutionary insight strengthens the translational relevance of our findings and supports the selection of these genes as meaningful targets for future therapeutic research.

These four compounds were chosen based on literature-supported neuroprotective and vasculotropic effects, and their structural non-redundancy was confirmed using Tanimoto similarity analysis, indicating their potential as functionally varied multi-target drugs. Moreover, each ligand was docked independently on a selected active site using Autodock Vina, and the results were analysed using Chimera and LigPlot. The results showed the best interaction with all six proteins selected. Amyloid beta precursor protein (APLP) has the best interaction with galangin, a well-known flavonoid that has neuroprotective properties and can change pathways associated with inflammation and oxidative stress, which is relevant to VaD [39]. These pharmacokinetic characteristics emphasize the translational potential of natural ligands in neurotherapeutics. While natural substances frequently have limited bioavailability, our data reveal that Curcumin, Galangin, Resveratrol, and Licocumarone have drug-like ADMET profiles similar to Donepezil and Memantine, two clinically used dementia therapies. This makes them viable scaffolds for further research or derivative use in future vascular dementia therapeutics.

Similarly, Claudin-5 [21,40] and superoxide dismutase [22,40] showed the best interaction with licocumarone, a coumarin derivative that has demonstrated vascular protective qualities that aid in the regulation of VaD [39]. Furthermore, Matrix metalloproteinase-9 [40] and Apolipoprotein E influence curcumin, which has potential therapeutic benefits in reducing oxidative stress and amyloid plaque formation in VaD due to its anti-inflammatory and antioxidant properties [32,41]. Moreover, Methylenetetrahydrofolate (MTHFR), specifically the C677T polymorphism, has been identified as a genetic factor associated with an increased risk of vascular dementia. [31] has a maximum number of interactions with resveratrol, and given its reputation for enhancing cerebral blood flow and reducing neuroinflammation, resveratrol is a promising candidate for vasodilation. Similarly, several underlying structural features can explain the variance in binding affinity between different protein-ligand complexes. Galangin's strong binding to APLP1 is attributed to its planar aromatic structure, which allows for π-π bonding and excellent hydrogen bonding with residues in the well-defined E2 domain pocket. The stiffness of this domain contributes to increased binding stability.

Curcumin, on the other hand, has a symmetrical and flexible polyphenolic structure that allows it to fit well into larger, hydrophobic clefts like MMP9 and APOE, generating stable connections via numerous hydrogen bonds and van der Waals forces. Resveratrol's smaller and more flexible structure fits perfectly into MTHFR's cofactor-binding domain, where its hydroxyl groups are predicted to interact with crucial polar residues involved in folate metabolism. Licocumarone's, coumarin backbone has modest affinity but creates stabilizing interactions with CLDN5 and SOD1, particularly in loop-rich, flexible areas. SOD1 and MMP9 have metal-coordinating centers ($Cu^{2+}$ and $Zn^{2+}$, respectively) that may help stabilize ligands through polar or coordination-like interactions. Collectively, these structural properties explain the observed docking scores and highlight the ligands' functional importance in the context of vascular dementia.

A group of thoroughly studied reference ligands and negative controls were docked against each chosen protein target to compare the docking results. These included Sodium Caprate for Claudin-5, Bexarotene for APOE, Betaine for MTHFR, Verubecestat for APLP1, SB-3CT for MMP9, and Edaravone for SOD1. In most cases, the natural compounds demonstrated superior or comparable binding affinities. For instance, Curcumin exhibited a stronger interaction with MMP9 (−8.0 kcal/mol) than SB-3CT (−7.2 kcal/mol), indicating its potential inhibitory effect. Similarly, Licocumarone bound SOD1 more effectively than Edaravone (−6.6 vs. −5.5 kcal/mol), suggesting possible antioxidant action. Curcumin also outperformed Bexarotene in APOE docking (−7.2 vs. −6.7 kcal/mol), reinforcing its role in lipid regulation and neuroprotection. Resveratrol showed a markedly stronger binding to MTHFR (−8.1 kcal/mol) compared to Betaine (−4.2 kcal/mol), indicating a greater capacity to modulate homocysteine metabolism. Galangin demonstrated strong binding to APLP1 (−8.5 kcal/mol), closely approaching that of the β-secretase inhibitor Verubecestat (−9.9 kcal/mol), suggesting relevance to amyloidogenic pathway regulation. While Sodium Caprate is functionally known to modulate Claudin-5, it exhibited a weaker score (−4.8 kcal/mol) than Licocumarone (−6.8 kcal/mol), highlighting the latter's potential BBB-modulatory role.

To further assess docking specificity, we also evaluated negative controls (Donepezil and Diclofenac) against the same targets. These compounds consistently showed weaker or nonspecific binding (mean∼−6.5 kcal/mol), confirming the selectivity of natural ligands toward disease-relevant proteins. The complete comparative data are summarized in Table 7.

To validate the docking results and evaluate the relative binding performance of the chosen natural ligands, we included clinically proven reference inhibitors (positive controls) and non-specific drugs (negative controls) for each protein target (see Table 7). The findings clearly show that natural ligands outperformed recognized inhibitors across numerous targets. For example, Curcumin has a higher binding score with MMP9 (−8.0 kcal/mol) than the reference inhibitor SB-3CT (−7.2 kcal/mol), indicating the possibility of competitive inhibition. Similarly, Licocumarone binds more firmly to SOD1 (−6.6 kcal/mol) than Edaravone (−5.5 kcal/mol), a clinically used antioxidant. Galangin has a high interaction with APLP1 (−8.5 kcal/mol), similar to the β-secretase inhibitor Verubecestat (−9.9 kcal/mol), indicating potential for regulating

**Table 7. Summarizes the comparison between the docking scores of protein targets with natural ligands, positive controls and negative controls.**

| Protein Target | Ligand | Type | Docking Score |
|---|---|---|---|
| APLP1 | Galangin | Natural | −8.5 |
| APLP1 | Verubecestat | Reference | −9.9 |
| APLP1 | Donepezil | Negative | −6.6 |
| CLDN5 | Licocumarone | Natural | −6.8 |
| CLDN5 | Sodium Caprate | Reference | −4.8 |
| CLDN5 | Diclofenac | Negative | −6.5 |
| MMP9 | Curcumin | Natural | −8.0 |
| MMP9 | SB-3CT | Reference | −7.2 |
| MMP9 | Donepezil | Negative | −6.6 |
| MTHFR | Resveratrol | Natural | −8.1 |
| MTHFR | Betaine | Reference | −4.2 |
| MTHFR | Diclofenac | Negative | −6.5 |
| SOD1 | Licocumarone | Natural | −6.6 |
| SOD1 | Edaravone | Reference | −5.5 |
| SOD1 | Donepezil | Negative | −6.6 |
| APOE | Curcumin | Natural | −7.2 |
| APOE | Bexarotene | Reference | −6.7 |
| APOE | Diclofenac | Negative | −6.5 |

amyloidogenic pathways. Negative controls like Donepezil and Diclofenac showed poorer binding (mean ~ −6.5 kcal/mol) across targets, supporting the selectivity and significance of natural ligands to VaD-associated proteins.

In addition, the docking results were validated by Normal Mode Analysis, through which several factors were analyzed, such as the B-factor, deformability, variance, and correlation. Each factor had a unique influence on the docked complexes, but the results showed a stable configuration for all targeted complexes. Furthermore, in previous studies, curcumin was found to have good therapeutic capability against vascular dementia; however, in this study, we deeply analyzed and targeted the most influential proteins and common natural compounds that showed positive results and could be used as a cost-effective and user-friendly treatment against this disease. While the current findings show substantial in-silico binding affinities and pathway involvement for natural chemicals across major VaD-related proteins, their larger significance stems from the potential for therapeutic translation. Given that targets such as APOE, MMP9, and CLDN5 are linked to not just vascular pathology but also early-stage cognitive decline and neurovascular disturbance, the discovered ligands could be used to develop early intervention techniques. Importantly, their multi-target binding patterns imply potential for polypharmacology, in which single drugs address numerous VaD characteristics at the same time, such as oxidative stress, amyloid dysregulation, and blood-brain barrier permeability. This is consistent with current dementia medication discovery trends, which favor multimodal therapy approaches.

Furthermore, because these compounds are natural products with acceptable safety profiles, they could be quickly moved to preclinical in vivo validation, particularly in vascular damage or chronic hypoperfusion models. Finally, the findings of this work support the use of structure-guided screening to hasten the identification of low-cost, mechanism-driven candidates for future clinical development in vascular dementia. While our findings provide useful computational insights, they are based solely on in-silico analyses and have not yet been validated experimentally. Further studies using cell or animal models are needed to confirm these ligand protein interactions and their biological effects. Several approaches are proposed to help translate computational insights into experimental validation. In vitro binding experiments, such as surface plasmon resonance (SPR) or fluorescence-based quenching studies, can be used to confirm the affinity of ligands for important targets such as APOE, MMP9, and APLP1. Enzyme activity assays can also determine compound's inhibitory potential against MMP9 or its ability to increase SOD1 antioxidant function. The effect of Claudin-5-targeting ligands on blood-brain barrier (BBB) integrity can be studied further utilizing cell-based permeability models, such as the hCMEC/D3 human brain endothelial monolayer system. For in vivo validation, well-established vascular dementia models, such as bilateral common carotid artery occlusion (BCCAO) in rats and chronic cerebral hypoperfusion in mice, can be used. These models enable for the evaluation of cognitive performance (e.g., Morris Water Maze test) and BBB integrity (e.g., Evans Blue dye extravasation assay) after administering candidate drugs such as Curcumin, Galangin, or Resveratrol. Together, these tests will give key molecular and functional insights, validating the therapeutic relevance of the natural ligands found in this study.

## Conclusion

This study introduces an innovative computational methodology integrating normal mode analysis with molecular docking to identify natural ligands possessing neuroprotective properties against vascular dementia. These results highlight the potential cost-effective therapeutic benefits of curcumin, galangin, licocumarone, and resveratrol through their strong interactions with significant neurodegenerative proteins. This study enhances our understanding of ligand-induced conformational alterations by uniquely integrating docking with evaluations of structural flexibility, in contrast to conventional drug discovery techniques. The observed strong binding affinities suggest that these natural compounds may regulate critical processes, such as amyloid metabolism, oxidative stress, and blood-brain barrier integrity, thereby mitigating the course of vascular dementia (VaD). This study employed a computational strategy to expedite drug discovery for neurodegeneration, effectively linking in-silico modelling with translational research. Validation of these interactions will advance the clinical use of these compounds. This study paves the way for targeted therapies for vascular dementia by harnessing the therapeutic potential of natural chemicals, thereby preventing the development of neuroprotective drugs.

## Supporting information

**S1 File.**
(DOCX)

## Author contributions

**Conceptualization:** Ailong Lin.

**Data curation:** Ailong Lin.

**Investigation:** Zhizhong Wang.

**Methodology:** Chunxian Wei, Ling Liu, Bizhou Bie.

**Project administration:** Chunxian Wei, Bizhou Bie.

**Software:** Zhiyong Li, Bizhou Bie.

**Supervision:** Zhiyong Li, Ling Liu, Bizhou Bie.

**Validation:** Sen Xu, Zhiyong Li.

**Visualization:** Zhiyong Li.

**Writing – original draft:** Zhizhong Wang, Yingchun Chen.

**Writing – review & editing:** Zhizhong Wang, Sen Xu.

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
