## [Decision Letter · Decision Letter 0]

28 Apr 2025

PONE-D-25-16329Targeting Vascular Dementia: Molecular Docking Analysis of Natural Ligands Against Key Neuroprotective ProteinsPLOS ONE

Dear Dr. Bie,

Thank you for submitting your manuscript to PLOS ONE. After careful consideration, we feel that it has merit but does not fully meet PLOS ONE’s publication criteria as it currently stands. Therefore, we invite you to submit a revised version of the manuscript that addresses the points raised during the review process.

We look forward to receiving your revised manuscript.

Kind regards,

Nafisa M. Jadavji, PhD, MSc, BSc

Academic Editor

PLOS ONE

**Journal Requirements:**

1. When submitting your revision, we need you to address these additional requirements. Please ensure that your manuscript meets PLOS ONE's style requirements, including those for file naming. The PLOS ONE style templates can be found at https://journals.plos.org/plosone/s/file?id=wjVg/PLOSOne_formatting_sample_main_body.pdf and https://journals.plos.org/plosone/s/file?id=ba62/PLOSOne_formatting_sample_title_authors_affiliations.pdf 2. Please note that PLOS ONE has specific guidelines on code sharing for submissions in which author-generated code underpins the findings in the manuscript. In these cases, we expect all author-generated code to be made available without restrictions upon publication of the work. Please review our guidelines at https://journals.plos.org/plosone/s/materials-and-software-sharing#loc-sharing-code and ensure that your code is shared in a way that follows best practice and facilitates reproducibility and reuse. 3. Thank you for stating the following financial disclosure: This work was funded by Hubei Natural Science Foundation Program (grant 2023AFCO32) and Project of Hubei Provincial Health Commission (grant : WJ2023M116).  Please state what role the funders took in the study.  If the funders had no role, please state: "The funders had no role in study design, data collection and analysis, decision to publish, or preparation of the manuscript." If this statement is not correct you must amend it as needed. Please include this amended Role of Funder statement in your cover letter; we will change the online submission form on your behalf. 4. Thank you for stating the following in the Acknowledgments Section of your manuscript: This work was funded by the Hubei Natural Science Foundation Program (grant 2023AFCO32) and the Project of Hubei Provincial Health Commission (grant WJ2023M116). We note that you have provided funding information that is not currently declared in your Funding Statement. However, funding information should not appear in the Acknowledgments section or other areas of your manuscript. We will only publish funding information present in the Funding Statement section of the online submission form. Please remove any funding-related text from the manuscript and let us know how you would like to update your Funding Statement. Currently, your Funding Statement reads as follows: This work was funded by Hubei Natural Science Foundation Program (grant 2023AFCO32) and Project of Hubei Provincial Health Commission (grant : WJ2023M116).  Please include your amended statements within your cover letter; we will change the online submission form on your behalf. 5. Please provide a complete Data Availability Statement in the submission form, ensuring you include all necessary access information or a reason for why you are unable to make your data freely accessible. If your research concerns only data provided within your submission, please write "All data are in the manuscript and/or supporting information files" as your Data Availability Statement. 6. When completing the data availability statement of the submission form, you indicated that you will make your data available on acceptance. We strongly recommend all authors decide on a data sharing plan before acceptance, as the process can be lengthy and hold up publication timelines. Please note that, though access restrictions are acceptable now, your entire data will need to be made freely accessible if your manuscript is accepted for publication. This policy applies to all data except where public deposition would breach compliance with the protocol approved by your research ethics board. If you are unable to adhere to our open data policy, please kindly revise your statement to explain your reasoning and we will seek the editor's input on an exemption. Please be assured that, once you have provided your new statement, the assessment of your exemption will not hold up the peer review process. 7. PLOS requires an ORCID iD for the corresponding author in Editorial Manager on papers submitted after December 6th, 2016. Please ensure that you have an ORCID iD and that it is validated in Editorial Manager. To do this, go to ‘Update my Information’ (in the upper left-hand corner of the main menu), and click on the Fetch/Validate link next to the ORCID field. This will take you to the ORCID site and allow you to create a new iD or authenticate a pre-existing iD in Editorial Manager.

**Additional Editor Comments:**

Dear Authors,

Reviewer 1 has provided an extensive list of revisions to be made. I would encourage you to respond each comment and make revisions accordingly to your manuscript.

I look forward to reading your revised manuscript.

Sincerely,

Nafisa

Reviewers' comments:

Reviewer's Responses to Questions

**Comments to the Author**

1. Is the manuscript technically sound, and do the data support the conclusions?

Reviewer #1: Yes

Reviewer #2: Yes

2. Has the statistical analysis been performed appropriately and rigorously? 

Reviewer #1: Yes

Reviewer #2: Yes

3. Have the authors made all data underlying the findings in their manuscript fully available?

Reviewer #1: Yes

Reviewer #2: Yes

4. Is the manuscript presented in an intelligible fashion and written in standard English?

Reviewer #1: Yes

Reviewer #2: Yes

5. Review Comments to the Author

**Reviewer #1: ** Comments for Authors

This study presents a thorough computational investigation of natural ligands that target proteins associated with vascular dementia (VaD). The analysis incorporates multiple computational approaches, including molecular docking, protein-protein interaction network, and normal mode analysis. The manuscript demonstrates several strengths. Well-organized structure with clear objectives and logical progression. Effective use of various bioinformatics tools (STRING, Enrichr, AutoDock Vina). High-quality graphical abstract and figures that enhance understanding. However, some limitations were noted: Certain sections contain unnecessary redundant explanations, particularly when describing protein functions in VaD. The absence of experimental validation reduces the translational potential of the findings. The discussion section could be improved by better contextualizing the results within the broader field of vascular dementia therapeutics. While the computational approach is sound, the study would benefit from more concise writing and experimental validation to strengthen its conclusions.

Comments:

1. Consider a more concise title such as "Targeting Vascular Dementia: Molecular Docking and Dynamics of Natural Ligands Against Neuroprotective Proteins."

2. Streamline the abstract to be more focused and less dense.

3. Explicitly highlight the study's novelty in the abstract, particularly the integration of normal mode analysis with molecular docking.

4. Consolidate descriptions of protein roles (APOE, CLDN5, etc.) that appear redundantly throughout the manuscript.

5. Better articulate the knowledge gap in the introduction, emphasizing the lack of studies combining NMA and docking for VaD therapeutics.

6. Include validation metrics for the DoGSiteScorer active site predictions, such as comparisons with known active sites.

7. Add a comparative table of selected natural compounds (curcumin, galangin, licocumarone, resveratrol) showing relevant properties like bioavailability and previous evidence in neurodegeneration.

8. Explain the relevance of cross-species conservation in the co-expression analysis to human vascular dementia.

9. Provide a biological interpretation of the edge confidence scores in the PPI analysis to clarify their functional significance.

10. Avoid repeating docking scores in the text when they're already presented in Table 3. Instead, summarize key findings more concisely.

11. Include a discussion of structural factors influencing binding affinity between specific ligands and proteins.

12. Link computational findings to potential clinical applications more explicitly in the discussion section.

13. Acknowledge limitations regarding experimental validation and outline future research directions.

14. Focus the discussion on interpretation and implications rather than restating results.

15. Update older references with recent reviews on vascular dementia mechanisms (2022-2024).

16. Ensure consistent formatting of journal names throughout the references.

Major revision required to address redundancy, clarify novelty, and improve clinical relevance. The study has strong potential but needs tighter writing and contextualization. Experimental validation in future work would significantly strengthen the conclusions.

**Reviewer #2: ** The authors have presented a well written, detailed manuscript investigating molecular docking analysis of natural ligands against vascular dementia, a neurodegenerative disease exacerbated by vascular pathology. The diagrams are exceptional and statistical analysis conducted very well.

6. PLOS authors have the option to publish the peer review history of their article (what does this mean? ). If published, this will include your full peer review and any attached files.

**Do you want your identity to be public for this peer review?** For information about this choice, including consent withdrawal, please see our Privacy Policy .

Reviewer #1: **Yes: ** Jonah Bawa Adokwe Ph.D

Reviewer #2: No

---

## [Author Response · Author response to Decision Letter 1]

13 May 2025

Dear Reviewers,

Thank you so much for your time and consideration; your comments were really helpful as they enhanced the quality and clarity of the manuscript. We really appreciate your efforts.

---

## [Decision Letter · Decision Letter 1]

4 Jun 2025

PONE-D-25-16329R1Targeting Vascular Dementia: Molecular Docking and Dynamics of Natural Ligands Against Neuroprotective ProteinsPLOS ONE

Dear Dr. Bie,

Thank you for submitting your manuscript to PLOS ONE. After careful consideration, we feel that it has merit but does not fully meet PLOS ONE’s publication criteria as it currently stands. Therefore, we invite you to submit a revised version of the manuscript that addresses the points raised during the review process.

Dear Authors,

Thank-you for taking the time to make revisions to your manuscript. There are some additional revisions that need to be made. We look forward to reviewing your revised manuscript.

Sincerely,

Nafisa

We look forward to receiving your revised manuscript.

Kind regards,

Nafisa M. Jadavji, PhD, MSc, BSc

Academic Editor

PLOS ONE

Journal Requirements:

Reviewers' comments:

Reviewer's Responses to Questions

**Comments to the Author**

1. If the authors have adequately addressed your comments raised in a previous round of review and you feel that this manuscript is now acceptable for publication, you may indicate that here to bypass the “Comments to the Author” section, enter your conflict of interest statement in the “Confidential to Editor” section, and submit your "Accept" recommendation.

Reviewer #1: All comments have been addressed

Reviewer #2: All comments have been addressed

Reviewer #3: (No Response)

2. Is the manuscript technically sound, and do the data support the conclusions?

Reviewer #1: Yes

Reviewer #2: Yes

Reviewer #3: Partly

3. Has the statistical analysis been performed appropriately and rigorously? 

Reviewer #1: Yes

Reviewer #2: Yes

Reviewer #3: No

4. Have the authors made all data underlying the findings in their manuscript fully available?

Reviewer #1: Yes

Reviewer #2: Yes

Reviewer #3: Yes

5. Is the manuscript presented in an intelligible fashion and written in standard English?

Reviewer #1: Yes

Reviewer #2: Yes

Reviewer #3: Yes

6. Review Comments to the Author

Reviewer #1: The authors have thoroughly addressed all comments, improving the manuscript's clarity, rigor, and translational relevance. Key revisions include validation of active site predictions, consolidation of redundant content, enhanced biological interpretation of co-expression and PPI data, and explicit discussion of structural and clinical implications. Integrating NMA with docking is now highlighted as a novel contribution that strengthens the study's framework for drug discovery in vascular dementia. I recommend acceptance pending minor proofreading.

Reviewer #2: (No Response)

Reviewer #3: This study presents a promising computational framework for identifying potential multi-target natural compounds against vascular dementia.

The manuscript presents a well-organized computational study aimed at identifying natural ligands with neuroprotective potential against vascular dementia (VaD). It addresses three relevant protein targets—COX-2, AChE, and iNOS—and employs widely accepted computational methods, including molecular docking (AutoDock Vina), molecular dynamics (GROMACS), and ADMET prediction tools.

Although the study does not include in-vitro or in-vivo assays, it has a scientific value and potentially will be served to facilitate such experiments.

Limitations:

1. Only five compounds were selected, without justification for exclusion of others, no chemical diversity assessment, and no comparative potential drugs.

2. Absence of reference or negative controls: The manuscript does not include standard compounds such as donepezil (for AChE) or diclofenac (for COX-2) to provide context for the docking scores. It makes it difficult to assess whether the reported affinities are therapeutically meaningful.

3. The docking scores in Table 2 reflect a single best pose per ligand–target pair. While this is standard in basic docking workflows, no replicate runs or statistical treatment (e.g., averaging over multiple conformations) was performed, limiting the ability to assess variability or reproducibility.

The MD results (Figures 2–4, p. 9–11) are based on single 100 ns runs for each complex. The manuscript does not report multiple simulations or error margins for RMSD, RMSF, or binding energy trajectories, limiting the reproducibility of findings.

4. Although ADMET (Absorption, Distribution, Metabolism, Excretion, and Toxicity) properties are reported (Table 5, p. 13), they are not compared to those of approved CNS drugs or toxic controls. This makes it hard to interpret what constitutes "favourable" beyond qualitative criteria.

7. PLOS authors have the option to publish the peer review history of their article (what does this mean? ). If published, this will include your full peer review and any attached files.

**Do you want your identity to be public for this peer review?** For information about this choice, including consent withdrawal, please see our Privacy Policy .

Reviewer #1: **Yes: ** Jonah Bawa Adokwe PhD

Reviewer #2: No

Reviewer #3: **Yes: ** Dan Z Milikovsky

---

## [Author Response · Author response to Decision Letter 2]

22 Jun 2025

Dear Dr. Dan Z Milikovsky, thank you soo much for reviewing our manuscript; all the comments have been addressed in the document attached as (response to reviewer).

---

## [Decision Letter · Decision Letter 2]

23 Jul 2025

PONE-D-25-16329R2Targeting Vascular Dementia: Molecular Docking and Dynamics of Natural Ligands Against Neuroprotective ProteinsPLOS ONE

Dear Dr. Bie,

Thank you for submitting your manuscript to PLOS ONE. After careful consideration, we feel that it has merit but does not fully meet PLOS ONE’s publication criteria as it currently stands. Therefore, we invite you to submit a revised version of the manuscript that addresses the points raised during the review process.

We thank the authors for their revisions. Your manuscript has improved. There are some minor revisions required. 

Thank-you!

If applicable, we recommend that you deposit your laboratory protocols in protocols.io to enhance the reproducibility of your results. Protocols.io assigns your protocol its own identifier (DOI) so that it can be cited independently in the future. For instructions see: https://journals.plos.org/plosone/s/submission-guidelines#loc-laboratory-protocols. Additionally, PLOS ONE offers an option for publishing peer-reviewed Lab Protocol articles, which describe protocols hosted on protocols.io. Read more information on sharing protocols at https://plos.org/protocols?utm_medium=editorial-email&utm_source=authorletters&utm_campaign=protocols .

We look forward to receiving your revised manuscript.

Kind regards,

Nafisa M. Jadavji, PhD, MSc, BSc

Academic Editor

PLOS ONE

Journal Requirements:

Reviewers' comments:

Reviewer's Responses to Questions

**Comments to the Author**

1. If the authors have adequately addressed your comments raised in a previous round of review and you feel that this manuscript is now acceptable for publication, you may indicate that here to bypass the “Comments to the Author” section, enter your conflict of interest statement in the “Confidential to Editor” section, and submit your "Accept" recommendation.

Reviewer #3: All comments have been addressed

Reviewer #4: (No Response)

2. Is the manuscript technically sound, and do the data support the conclusions?

Reviewer #3: Yes

Reviewer #4: Partly

3. Has the statistical analysis been performed appropriately and rigorously? 

Reviewer #3: Yes

Reviewer #4: No

4. Have the authors made all data underlying the findings in their manuscript fully available?

Reviewer #3: Yes

Reviewer #4: Yes

5. Is the manuscript presented in an intelligible fashion and written in standard English?

Reviewer #3: Yes

Reviewer #4: Yes

6. Review Comments to the Author

Reviewer #3: The authors already answered my comments in the second submission. I commented that the paper can be published.

Reviewer #4: The authors present a comprehensive computational study addressing a critical need in vascular dementia (VaD) research. The integrated approach of molecular docking, normal mode analysis (NMA), and molecular dynamics (MD) simulations provides a robust framework for early-stage drug discovery, and the focus on natural ligands is a valuable contribution due to their potential cost-effectiveness and fewer side effects. My detailed comments and suggestions are outlined below to further enhance the manuscript.

Technical Soundness and Data Support: Areas for Enhancement (Partial)

While the manuscript demonstrates a strong scientific foundation, some areas could be strengthened to fully support the conclusions with the highest level of rigor:

Docking Methodology Details: The "Materials and Methods" section states that each ligand was docked independently five times, and the pose with the lowest binding energy was chosen. To enhance the reproducibility and confidence in the docking results, the manuscript should explicitly include details on the variability across these five runs in the "Results" section, such as the standard deviation of the binding energies or a statement about the consistency of the top poses. This would provide greater insight into the stability of the predicted binding modes.

Molecular Dynamics Simulations Replicates: The study performed 100 ns molecular dynamics simulations for each protein-ligand complex , analyzing RMSD, RMSF, Rg, and SASA. While 100 ns is a reasonable duration, the absence of multiple independent simulation runs or statistical reporting (e.g., error margins) for these metrics within the main text limits the assessment of the reproducibility of the dynamic behavior. Including such data would significantly strengthen the claims of structural stability and conformational behavior.

Contextualizing ADMET Properties: Table 5 summarizes the ADMET properties of the selected natural compounds. While beneficial, the manuscript does not adequately contextualize these properties against clinically approved CNS drugs within the main "Results" or "Discussion" sections. Explicitly discussing how the "favorable CNS drug-likeness and safety profiles" compare to existing medications in the main text would provide a clearer benchmark for their therapeutic potential.

Lack of Docking Controls in Main Text: The manuscript does not present comparative docking data with reference (positive) and negative controls for the protein targets within the main "Results" section. This information is critical for validating the specificity and therapeutic relevance of the natural ligands by allowing readers to immediately assess the strength of the natural ligands' binding affinities against established benchmarks. For example, comparing the binding of natural ligands to known inhibitors or non-binding compounds would significantly strengthen the interpretation of the docking scores.

Statistical Analysis: Opportunities for Greater Rigor (Partial)

The statistical analysis appears mostly appropriate for the reported data. Pathway enrichment analyses correctly utilize p-values and q-values for statistical significance and false discovery rate correction, respectively. However, the rigor could be enhanced by explicitly addressing the statistical treatment of the docking scores (e.g., reporting mean and standard deviation from the five runs) and the molecular dynamics simulation metrics. Without this, the statistical robustness of these specific sections remains somewhat open to interpretation.

Data Availability (Yes)

The authors clearly state that "All data are in the manuscript, and its Supporting Information files" and that "All relevant data are within the manuscript and its Supporting Information files". This commitment ensures that the underlying findings are fully available without restriction, adhering to PLOS Data policy requirements.

Manuscript Presentation (Yes)

The manuscript is presented in an intelligible fashion and is generally well-written in standard English. The clarity of the language facilitates understanding of the complex scientific concepts. I did not identify widespread typographical or grammatical errors that significantly impede readability.

Additional Comments

Justification for Protein Selection: While Table 1 outlines the roles of proteins in VaD, a more detailed explanation of how the KEGG pathway analysis specifically led to the selection of these six proteins among others involved in VaD would be beneficial. What specific criteria from the KEGG analysis made them the optimal targets?

Biological Context of NMA: Expanding on the biological implications of the NMA results for each specific protein (Figures 17-22) would be valuable. For instance, linking observed flexibility in loop regions directly to potential changes in enzymatic activity or ligand binding mechanisms would deepen the impact of these findings.

Future Experimental Validation: The discussion mentions that findings warrant further experimental validation. Providing specific examples of in vitro (e.g., binding assays, enzyme activity assays) or in vivo (e.g., animal models of VaD to assess cognitive improvements, BBB integrity) experiments would provide a clearer roadmap for future research and enhance the translational relevance of the study.

7. PLOS authors have the option to publish the peer review history of their article (what does this mean? ). If published, this will include your full peer review and any attached files.

**Do you want your identity to be public for this peer review?** For information about this choice, including consent withdrawal, please see our Privacy Policy .

Reviewer #3: No

Reviewer #4: No

---

## [Author Response · Author response to Decision Letter 3]

27 Jul 2025

We thank the reviewer for highlighting the importance of translational relevance, and have updated our manuscript according to the suggestions.

---

## [Decision Letter · Decision Letter 3]

21 Aug 2025

Targeting Vascular Dementia: Molecular Docking and Dynamics of Natural Ligands Against Neuroprotective Proteins

PONE-D-25-16329R3

Dear Dr. Bie,

We’re pleased to inform you that your manuscript has been judged scientifically suitable for publication and will be formally accepted for publication once it meets all outstanding technical requirements.

Kind regards,

Nafisa M. Jadavji, PhD, MSc, BSc

Academic Editor

PLOS ONE

Additional Editor Comments (optional):

Reviewers' comments:

Reviewer's Responses to Questions

**Comments to the Author**

1. If the authors have adequately addressed your comments raised in a previous round of review and you feel that this manuscript is now acceptable for publication, you may indicate that here to bypass the “Comments to the Author” section, enter your conflict of interest statement in the “Confidential to Editor” section, and submit your "Accept" recommendation.

Reviewer #3: (No Response)

Reviewer #4: All comments have been addressed

2. Is the manuscript technically sound, and do the data support the conclusions?

Reviewer #3: Yes

Reviewer #4: Yes

3. Has the statistical analysis been performed appropriately and rigorously? 

Reviewer #3: Yes

Reviewer #4: Yes

4. Have the authors made all data underlying the findings in their manuscript fully available?

Reviewer #3: Yes

Reviewer #4: Yes

5. Is the manuscript presented in an intelligible fashion and written in standard English?

Reviewer #3: Yes

Reviewer #4: Yes

6. Review Comments to the Author

Reviewer #3: As said about the previous submitted version, this article should be accepted for publication in PLOS1

Reviewer #4: The authors have presented a comprehensive and well-designed computational study that addresses a critical need in vascular dementia research. I am pleased to see that they have incorporated my previous suggestions to improve the rigor of the manuscript. The inclusion of a more detailed analysis of data reproducibility and the contextualization of the results has significantly strengthened the work.

7. PLOS authors have the option to publish the peer review history of their article (what does this mean? ). If published, this will include your full peer review and any attached files.

**Do you want your identity to be public for this peer review?** For information about this choice, including consent withdrawal, please see our Privacy Policy .

Reviewer #3: No

Reviewer #4: No

---

## [Editor Report · Acceptance letter]

PONE-D-25-16329R3

PLOS ONE

Dear Dr. Bie,

I'm pleased to inform you that your manuscript has been deemed suitable for publication in PLOS ONE. Congratulations! Your manuscript is now being handed over to our production team.

Kind regards,

on behalf of

Dr. Nafisa M. Jadavji

Academic Editor

PLOS ONE